# GeoDT: Geometry-Inspired Decision Transformer for Robust Safe Multi-Task Offline Reinforcement Learning

## Abstract

Scaling offline reinforcement learning across heterogeneous tasks remains challenging, especially under safety constraints. In multi-task settings, features processed by a shared model may play different semantic roles across tasks, leading to semantic inconsistency, conflicting optimization signals, and performance degradation as task diversity increases. Existing multi-task and safe offline RL methods address parts of this problem, but few jointly address semantic inconsistency and safety when a single policy is trained across heterogeneous tasks. We propose GeoDT (Geometry-Inspired Decision Transformer), a framework for safe multi-task offline RL that biases cross-task sharing toward transition-related, agent-centric trajectory structure. Here, we use "geometry" in an operational sense to denote spatial and motion-related structure in agent trajectories and their action-conditioned transition regularities. GeoDT learns a soft, data-driven decomposition that biases one representation branch toward transition-predictive information while retaining complementary task-dependent information in the other branch. It constructs structured context from prompt trajectories through relational structure induction and prototype memory, and incorporates safety by using cost signals to reweight trajectory relations and retrieved patterns according to feasibility. The resulting context is fused with complementary task-dependent features to condition a cost-aware Decision Transformer. To evaluate robustness as task diversity increases, we introduce the Task Scaling Robustness Score (TSRS) and Inter-Task Balance Score (ITBS), which measure performance retention and cross-task balance. Experiments on safe multi-task offline RL benchmarks show that GeoDT achieves strong reward–cost trade-offs, improved robustness under increasing task diversity, and zero-shot adaptation to unseen safety budgets. These results suggest that transition-related, agent-centric trajectory structure can provide an effective basis for safe multi-task offline reinforcement learning.

## 1 Introduction

Real-world decision-making systems, from robotic control to safety-critical automation, increasingly require agents that can operate across multiple tasks while respecting explicit safety constraints (Wu et al., 2024). Offline reinforcement learning (RL) is a natural framework for such settings, since it learns from fixed datasets without unsafe online exploration. However, learning a single offline policy across heterogeneous tasks remains challenging, especially when a single policy must jointly handle varying objectives, contextual observations, and safety requirements.

A key obstacle in heterogeneous multi-task offline RL is semantic inconsistency under shared representation learning, where a common encoder must process features whose semantic roles may differ across tasks, often leading to task interference or negative transfer (Yu et al., 2021; Ishfaq et al., 2024). In heterogeneous multi-task settings, observations from different tasks are often aligned into a shared input format and processed by a common encoder, but aligned input dimensionality does not imply aligned semantics. Features handled by the same model can still correspond to different physical quantities, contextual cues, or safety-relevant signals across tasks. As a result, naive parameter sharing can induce representation conflict, conflicting optimization signals, and negative transfer as task diversity increases.

This challenge is particularly pronounced in the offline safe RL setting. Because training is restricted to static datasets, the agent cannot rely on additional exploration to disambiguate misleading shared representations or recover from harmful cross-task interference, while distributional shift further amplifies errors under joint sharing (Fujimoto et al., 2019). At the same time, safety introduces an additional layer of difficulty: beyond learning task-relevant behavior, the agent must also determine which trajectory patterns remain feasible under task-dependent cost budgets (Lin et al., 2023). In other words, multi-task offline safe RL requires not only effective cross-task sharing, but also selective sharing that respects safety. Existing approaches address only parts of this problem. Multi-task offline RL methods often reduce interference through task identifiers, task embeddings, or modular conditioning (Yang et al., 2020; Hu et al., 2024; Yu et al., 2021; Ishfaq et al., 2024), but such approaches either rely on explicit task identity or do not specify which part of a semantically mixed state should be shared, leaving transition-related agent dynamics entangled with task-dependent environmental cues. Prompt- and context-based policy learning methods improve adaptation and transfer (Laskin et al., 2022; Huang et al., 2024; Yoo et al., 2025), yet they primarily focus on performance generalization rather than semantic misalignment under joint training. Meanwhile, safe offline RL methods have shown strong results in single-task settings (Zheng et al., 2024; Koirala et al., 2024b; Yao et al., 2024; Gong et al., 2025; Chemingui et al., 2025b; Liu et al., 2023b; Zhang et al., 2023), but they do not explicitly address the representation conflict introduced by heterogeneous multi-task learning. As a result, few existing methods jointly tackle semantic inconsistency, robustness under task diversity, and safety in a unified offline multi-task framework.

We argue that a more stable basis for cross-task sharing lies in the *transition-related, agent-centric structure* of trajectories. Although task-specific semantics, reward functions, and contextual cues may vary substantially, the agent's own motion and interaction with the environment often exhibit reusable regularities across tasks. We use *geometry-inspired* in an operational sense to refer to such spatial and motion-related trajectory structure together with its action-conditioned transition regularities. We use the term in this operational sense throughout the paper. Such agent-centric structure offers a more reliable anchor for cross-task sharing than raw task semantics alone. Moreover, safety should not be treated merely as a decision-time constraint appended after representation learning; it should also shape which trajectory relations and structural patterns are feasible to reuse across tasks.

Motivated by this view, we propose **GeoDT** (Geometry-Inspired Decision Transformer), a framework for safe multi-task offline RL. GeoDT employs a geometry-inspired decomposition module that softly routes state information into a transition-related branch and a complementary task-dependent branch. A self-supervised one-step consistency objective regularizes the transition-related branch, which is subsequently used to construct structured context through relational structure induction and prototype memory. To incorporate safety without entangling it with the representation itself, GeoDT uses cost information to modulate which relational patterns and structural memories are emphasized, so that feasibility shapes the reuse of transition-related structure across tasks. The resulting context is fused with complementary task-dependent features to condition a cost-aware Decision Transformer (Chen et al., 2021), enabling robust and safety-aware policy learning without explicit task identifiers. To further support this motivation, we provide a simplified theoretical analysis and a targeted empirical gradient analysis in Appendix B, illustrating how semantic inconsistency under shared training can induce cross-task optimization difficulties and how a transition-biased decomposition can help alleviate them.

Meanwhile, standard average reward and cost can hide two failure modes that are central to multi-task scaling: substantial degradation relative to single-task reference performance, and uneven performance concentrated on only a subset of tasks. To better assess robustness as task diversity increases, we further introduce two diagnostic metrics: the Task Scaling Robustness Score (TSRS) and the Inter-Task Balance Score (ITBS). TSRS and ITBS are designed to capture these two aspects, namely performance retention and cross-task balance, as the number of jointly trained tasks increases.

Our contributions are summarized as follows:

- We identify semantic inconsistency under shared encoding as a key obstacle in safe multi-task offline RL with heterogeneous tasks, and formalize a practical setting where tasks differ in objectives,

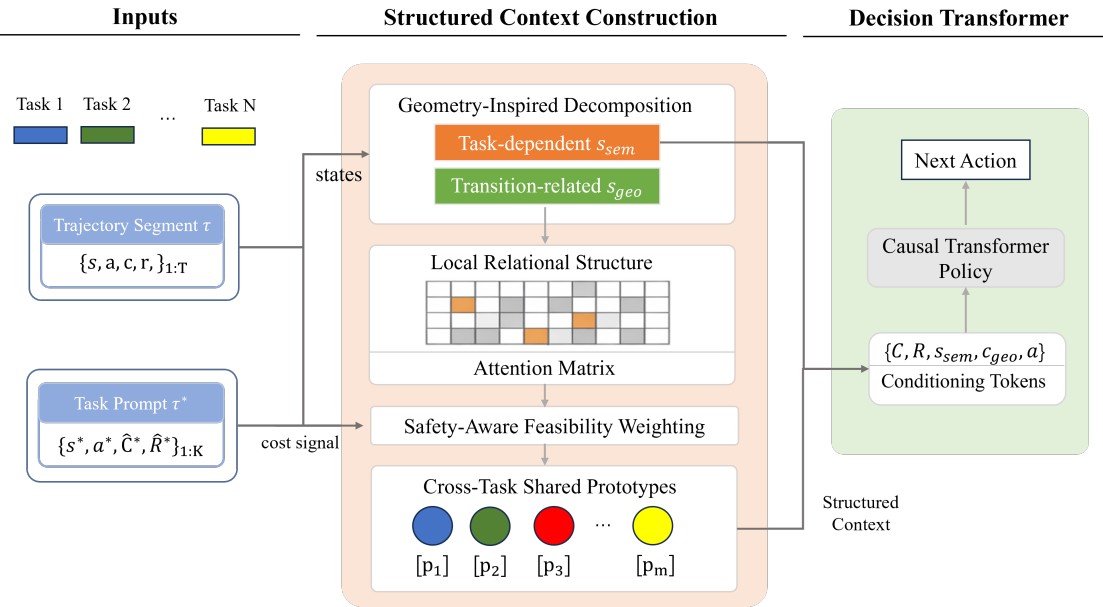

Figure 1: Overview of GeoDT. To mitigate semantic inconsistency across heterogeneous tasks, GeoDT employs a geometry-inspired decomposition module that softly routes state information into a transition-related branch and a complementary task-dependent branch. A prompt trajectory $\tau^\star$ is used to build the structured context $c_{geo}$ through relational structure induction, safety-aware feasibility reweighting, and shared prototype retrieval. The resulting context is fused with complementary task-dependent state features and fed into a cost-aware Decision Transformer for reward- and cost-conditioned action prediction.

contextual semantics, and safety constraints, with additional zero-shot evaluation under varying target safety budgets.

- We propose GeoDT, a geometry-inspired Decision Transformer that biases cross-task sharing toward transition-related, agent-centric trajectory structure through a geometry-inspired decomposition, self-supervised consistency learning, relational structure induction, prototype memory, and safety-aware feasibility reweighting.

- We introduce TSRS and ITBS as diagnostic measures of robustness under increasing task diversity, and conduct extensive experiments demonstrating strong reward–cost trade-offs, robustness under increasing task diversity, and zero-shot generalization to unseen safety budgets.

The remainder of this paper is organized as follows. Section 2 reviews related work on safe offline RL, multi-task offline RL, and structure-aware sequence modeling. Section 3 introduces the problem formulation for multi-task safe offline RL. Section 4 presents GeoDT, including geometry-inspired decomposition, relational structure induction, prototype memory, and safety-aware decision conditioning. Section 5 describes the experimental setup and empirical results, including robustness evaluation under increasing task diversity and ablation studies. Finally, Section 6 concludes with limitations and future directions.

## 2 Related Work

**Safe offline reinforcement learning.** Safe offline reinforcement learning studies how to learn policies from fixed datasets while satisfying safety constraints. Early work introduced constrained batch policy optimization and conservative policy learning under offline settings (Le et al., 2019; Fujimoto et al., 2019). Subsequent approaches improved safety through policy projection (Polosky et al., 2022), value regularization against unsafe out-of-distribution actions (Xu et al., 2022a), and stationary-distribution correction under cost constraints (Lee et al., 2022). More recent methods have adopted generative and sequence-modeling

paradigms. CDT (Liu et al., 2023b) and SaFormer (Zhang et al., 2023) extend Transformer-based decision modeling to constrained settings, while FISOR (Zheng et al., 2024), OASIS (Yao et al., 2024), LSPC (Koirala et al., 2024b), FAWAC (Koirala et al., 2024a), TraC (Gong et al., 2025), and CAPS (Chemingui et al., 2025b) improve reward–cost trade-offs through diffusion modeling, latent-space optimization, feasibility-aware regression, unsafe-trajectory filtering, or adaptive constraint handling. These methods have demonstrated strong performance in single-task safe offline RL, but most assume fixed task semantics and do not address the additional representation conflict that arises when heterogeneous tasks are trained jointly.

**Multi-task and context-based offline reinforcement learning.** Offline multi-task RL aims to learn a shared policy or sequence model across multiple tasks from static data. Existing methods mitigate task interference through sample routing, factorized representations, or task-conditioned parameterization. Conservative data sharing (Yu et al., 2021) selectively routes transitions to reduce negative transfer, while MORL (Ishfaq et al., 2024) learns low-rank shared structure across tasks. Recent multi-task Decision Transformer methods further manage task heterogeneity through parameter specialization. HarmoDT (Hu et al., 2024) learns task-specific parameter masks that define a subspace for each task, whereas M3DT (Kong et al., 2025) augments a Prompt-DT backbone with mixture-of-experts modules and grouped expert training to scale to large task sets. Prompt-conditioned methods provide an alternative way to infer task information from demonstrations. Prompt-DT (Xu et al., 2022b) prepends short trajectory segments that encode task-specific behavior and support few-shot adaptation without policy fine-tuning. In the safe setting, CoPDT (Xue et al., 2026) introduces a constraint-prioritized prompt encoder that uses sparse cost signals to identify constraint-relevant information under varying safety budgets. In its cross-environment setting, however, CoPDT uses environment-specific state and action encoders selected according to an environment identity. Other approaches improve cross-task generalization through task-dependent architectures, latent skill abstractions, and context-conditioned sequence modeling (Yang et al., 2020; Yoo et al., 2022; Laskin et al., 2022; Lee et al., 2023; Wang et al., 2025; Yoo et al., 2025). However, most primarily target performance transfer and do not explicitly address the semantic inconsistency that arises when heterogeneous observations are aligned into a shared representation but retain different semantic roles across tasks. Moreover, safety constraints are often absent or handled only implicitly.

**Structure-aware and dynamics-aware trajectory modeling.** A growing line of work suggests that effective generalization in sequential decision making depends not only on task identity, but also on how structural information is extracted from trajectories and interactions. Context-based sequence models use prompts or retrieved histories to infer task-relevant behavior patterns (Laskin et al., 2022; Xu et al., 2022b; Huang et al., 2024; Yoo et al., 2025), while representation-learning methods for control seek to separate shared dynamical structure from task-specific variation through latent abstractions, temporal consistency, or structured memory (Gelada et al., 2019; Schwarzer et al., 2020; Khetarpal et al., 2024). Geometry-aware methods have also used explicit object geometry, geometric reasoning, or equivariant spatial priors to improve generalization (Bellemare et al., 2019; Hoang et al., 2025; Gao et al., 2025; Cai et al., 2024; Li et al., 2025).

Our work follows this broader structure-aware perspective but differs in three respects. First, we study heterogeneous multi-task offline RL, where semantic inconsistency under shared encoding is a central challenge. Second, we use agent-centric, transition-related trajectory structure as the basis for cross-task sharing, rather than explicit task identifiers or hand-specified geometric priors. Third, we use safety signals to reweight which trajectory relations and structural patterns are emphasized for reuse. Accordingly, "geometry" is used here in an operational, geometry-inspired sense, rather than to denote explicit geometric reasoning or equivariant modeling.

## 3 Preliminaries

### 3.1 Constrained Markov Decision Processes and Safe Multi-Task Offline RL

Safe reinforcement learning is commonly formulated under the constrained Markov decision process (CMDP) framework (Altman, 2021). A CMDP is defined as

$$\mathcal{M} = (\mathcal{S}, \mathcal{A}, P, r, c, \gamma),$$

where $\mathcal{S}$ is the state space, $\mathcal{A}$ is the action space, $P : \mathcal{S} \times \mathcal{A} \to \mathcal{P}(\mathcal{S})$ is the transition distribution, $r : \mathcal{S} \times \mathcal{A} \to \mathbb{R}$ is the reward function, $c : \mathcal{S} \times \mathcal{A} \to \mathbb{R}_{\geq 0}$ is the safety cost, and $\gamma \in [0, 1)$ is the discount factor. The goal is to learn a policy $\pi$ that maximizes the expected discounted return while keeping the expected cumulative cost below a budget $\kappa$:

$$\max_{\pi} \; \mathbb{E}_{\tau \sim \pi}[R(\tau)] \qquad \text{s.t.} \qquad \mathbb{E}_{\tau \sim \pi}[C(\tau)] \leq \kappa, \tag{1}$$

where $R(\tau) = \sum_{t=1}^{T} \gamma^{t-1} r_t$ and $C(\tau) = \sum_{t=1}^{T} \gamma^{t-1} c_t$. In this work, we consider a *multi-task safe offline RL* setting with a collection of $N$ tasks, indexed by $i \in \{1, \ldots, N\}$:

$$\{\mathcal{M}_i\}_{i=1}^{N}, \qquad \mathcal{M}_i = (\mathcal{S}_i, \mathcal{A}, P_i, r_i, c_i, \gamma).$$

The tasks share the same action space $\mathcal{A}$ but may differ in state semantics, transition dynamics, reward functions, and cost functions. In particular, even when state vectors are represented in a shared input format or have the same dimensionality across tasks, corresponding features may encode different physical meanings, giving rise to semantic inconsistency and cross-task interference during joint training.

For each task, we assume access only to an offline dataset collected by unknown behavior policies, with no additional environment interaction during training. Let $\mathcal{D} = \bigcup_{i=1}^{N} \mathcal{D}_i$ denote the union of task datasets, where each transition sequence in $\mathcal{D}_i$ consists of tuples $(s_t, a_t, r_t, c_t, s_{t+1})$. At test time, the agent is required to produce actions that achieve favorable reward–cost trade-offs under a specified target safety budget, while generalizing across heterogeneous tasks without explicit task identifiers.

Our goal is to learn a single offline policy for heterogeneous safe RL tasks that remains robust as task diversity increases and is adaptive to different safety budgets. The main challenge is that naive parameter sharing across tasks can be ineffective when state dimensions are semantically misaligned. GeoDT addresses this challenge by identifying transition-related, agent-centric structure that is more stable across tasks, using it to construct structured context for action prediction, and incorporating safety as a signal that influences which shared structural patterns are feasible to reuse under a given budget.

### 3.2 Decision Transformer

Decision Transformer (DT) (Chen et al., 2021) formulates offline RL as a sequence modeling problem. Instead of learning value functions explicitly, DT models actions autoregressively conditioned on trajectory context and desired outcomes using a causal Transformer. A standard DT takes as input an interleaved sequence

$$(\hat{R}_1, s_1, a_1, \hat{R}_2, s_2, a_2, \ldots),$$

where $\hat{R}_t = \sum_{i=t}^{T} r_i$ denotes the return-to-go at step $t$. Given a finite context window, the model predicts the next action by

$$a_t = f_\theta(\hat{R}_{t-L:t}, \, s_{t-L:t}, \, a_{t-L:t-1}),$$

where $f_\theta$ is implemented as a causal Transformer decoder. Several works extend Decision Transformer with richer conditioning signals. Prompt-DT (Xu et al., 2022b) prepends a short trajectory prompt to the input sequence as an implicit task descriptor, enabling multi-task learning and few-shot adaptation without directly providing a task identifier to the policy. In safe offline RL, CDT (Liu et al., 2023b) and SaFormer (Zhang et al., 2023) incorporate safety-related conditioning signals, including cost-to-go, into sequence-based action prediction. GeoDT inherits the causal DT backbone, return-cost conditioning, and trajectory-prompt paradigm. Its main distinction is that, instead of conditioning directly on raw prompt tokens, it constructs a structured context $c_{\text{geo}}$ through geometry-inspired decomposition, relational structure induction, feasibility-aware reweighting, and prototype retrieval for joint learning across semantically heterogeneous safe tasks. The resulting context is fused with complementary task-dependent state features, as detailed in Section 4.

## 4 Method

Figure 1 illustrates the overall architecture of **GeoDT**. GeoDT does not provide an explicit task or environment identifier to the policy network. Instead, it uses a short trajectory prompt sampled from the offline data as

compact context about the current task behavior and safety conditions. A prompt associated with task $\mathcal{T}i$ is defined as

$$\tau_i^\star = \left( \hat{R}_{t:t+K}^\star, \hat{C}_{t:t+K}^\star, s_{t:t+K}^\star, a_{t:t+K}^\star \right), \tag{2}$$

where $\hat{R}_t^\star = \sum_{j=t}^T r_j$, $\hat{C}_t^\star = \sum_{j=t}^T c_j$, and $K$ is the prompt length. Because $K$ is substantially shorter than the full trajectory horizon, $\tau_i^\star$ provides a compact partial-trajectory context rather than a complete demonstration.

**Terminology and notation.** As introduced in Section 1, we use "geometry" in a geometry-inspired, operational sense. We decompose each state into $s_t^{\text{geo}}$ and $s_t^{\text{sem}}$. The former is regularized toward action-conditioned transition predictability and is intended to retain information associated with agent-centric trajectory structure. The latter is defined through the complementary routing mask; it is not explicitly supervised to encode semantics, but is intended to preserve complementary task-dependent information. We denote the structured prompt context constructed from the transition-predictive branch by $c_{\text{geo}}$. We emphasize that this decomposition is soft and preferential rather than a hard or information-pure disentanglement. GeoDT does not impose explicit Euclidean constraints, equivariance, or symbolic geometric structure. Accordingly, the subscript "geo" denotes the intended geometry-inspired, transition-related inductive bias rather than an explicit geometric guarantee.

**End-to-end overview.** GeoDT is based on the principle that trajectory prompts contain both reusable transition-related structure and complementary task-dependent information, which should play different roles in cross-task sharing. The full pipeline proceeds in four steps. First, a geometry-inspired decomposition module softly routes each state into a transition-related component $s_t^{\text{geo}}$ and a complementary component $s_t^{\text{sem}}$, while a one-step consistency objective regularizes the former (Section 4.1). Second, GeoDT induces relational structure among timesteps within the prompt. Third, cost-to-go signals reweight these relations according to feasibility, and the resulting prompt representation retrieves structural patterns from a shared prototype memory to form the context $c_{\text{geo}}$ (Section 4.2). Finally, $c_{\text{geo}}$ is fused with the complementary state features and passed, together with return-to-go and cost-to-go, to a causal Decision Transformer for action prediction (Section 4.3). Algorithm 1 summarizes training, while Algorithm 2 in Appendix C summarizes the complete training and deployment procedures.

### 4.1 Geometry-Inspired Decomposition and Consistency Learning

A central challenge in heterogeneous multi-task offline RL is *semantic inconsistency*: even when observations are aligned into a common input format, corresponding dimensions may encode different meanings across tasks, leading to conflicting optimization signals under shared training. To mitigate this issue, GeoDT employs a geometry-inspired decomposition module that softly routes each state into two complementary components: a transition-related branch and a task-dependent branch.

Given a state $s_t \in \mathbb{R}^d$, we compute a state-dependent gating vector

$$m_t = \sigma(\text{MLP}(s_t)) \in (0,1)^d, \tag{3}$$

and define

$$s_t^{\text{geo}} = m_t \odot s_t, \qquad s_t^{\text{sem}} = (1 - m_t) \odot s_t, \tag{4}$$

where $\odot$ denotes element-wise multiplication. Because $m_t$ is continuous and state-dependent, this mechanism performs soft, data-driven routing rather than a hard partition, and does not require an explicit task identifier.

To provide a transition-based criterion for learning the routing mask, we introduce an auxiliary one-step consistency objective. A predictor $f_\psi$ estimates the next routed component as

$$\hat{s}_{t+1}^{\text{geo}} = f_\psi(s_t^{\text{geo}}, a_t), \tag{5}$$

and is trained using

$$\mathcal{L}_{\text{geo}} = \mathbb{E}_{(s_t, a_t, s_{t+1}) \sim \mathcal{D}} \left[ \left\| f_\psi(s_t^{\text{geo}}, a_t) - s_{t+1}^{\text{geo}} \right\|_2^2 \right]. \tag{6}$$

Although $\mathcal{L}_{\mathrm{geo}}$ takes the form of an auxiliary one-step prediction loss, its role in GeoDT is to provide a transition-based criterion for learning the soft routing mask. Specifically, it biases $s_t^{\mathrm{geo}}$ toward action-conditioned, transition-predictive information that can be used for structured context construction. The loss does not enforce exact invariance or an information-pure factorization; rather, it supplies an inductive bias that complements the policy objective.

The component $s_t^{\mathrm{sem}}$ is designed to retain complementary task-dependent information that is not preferentially routed to $s_t^{\mathrm{geo}}$. However, it is defined only through the complementary mask and is not explicitly supervised with semantic labels. We therefore use the superscript "sem" as a functional shorthand rather than as a claim of information-pure semantic disentanglement. In the following, $s_t^{\mathrm{geo}}$ is used to construct the structured prompt context, whereas $s_t^{\mathrm{sem}}$ provides complementary task-dependent features for policy conditioning. Appendix B provides a simplified analysis and empirical gradient measurements illustrating how transition-biased routing can reduce optimization conflict under heterogeneous shared training.

## 4.2 Structured Context Construction

Given a prompt trajectory $\tau^\star$, we map its transition-related component into a latent embedding space:

$$z_u^{\mathrm{geo}} = \phi(s_u^{\mathrm{geo}}) \in \mathbb{R}^{d_h}, \tag{7}$$

where $u \in \{1, \ldots, K\}$ indexes a timestep within the prompt and $d_h$ is the latent dimension. For clarity, we omit the prompt superscript $\star$ from individual tokens in this subsection.

**Relational structure induction.** We model dependencies among timesteps within the same prompt using data-dependent pairwise attention:

$$\widetilde{A}_{uv} = \frac{(W_q z_u^{\mathrm{geo}})^\top (W_k z_v^{\mathrm{geo}})}{\sqrt{d_h}}, \qquad A_{uv} = \mathrm{softmax}_v\left(\widetilde{A}_{uv}\right), \tag{8}$$

where $u$ and $v$ index positions in a single prompt, rather than in different prompts or tasks. The corresponding prompt representation is

$$h_{\mathrm{prompt}} = \mathrm{Pool}\left(\left\{\sum_{v=1}^{K} A_{uv} W_v z_v^{\mathrm{geo}}\right\}_{u=1}^{K}\right), \tag{9}$$

where $\mathrm{Pool}(\cdot)$ denotes mean pooling over the $K$ prompt positions. Although this attention is computed within one prompt, its parameters are jointly learned from all task datasets.

**Safety-aware structure reweighting.** We use cost-to-go as a feasibility signal to reweight prompt relations. For each prompt position, we define a feasibility weight

$$w_u = \sigma\left(-\alpha \hat{C}_u\right), \qquad A_{uv}^{\mathrm{safe}} = \mathrm{softmax}_v\left(w_u w_v \widetilde{A}_{uv}\right), \tag{10}$$

where $\alpha > 0$ controls the strength of the feasibility modulation. We then obtain the safety-aware prompt summary by replacing $A_{uv}$ with $A_{uv}^{\mathrm{safe}}$ in the aggregation above:

$$h_{\mathrm{prompt}}^{\mathrm{safe}} = \mathrm{Pool}\left(\left\{\sum_{v=1}^{K} A_{uv}^{\mathrm{safe}} W_v z_v^{\mathrm{geo}}\right\}_{u=1}^{K}\right). \tag{11}$$

Thus, cost information affects which prompt relations are emphasized, without directly modifying the routed state representation.

**Prototype-based global context.** To complement the local prompt structure with recurring patterns shared across tasks, we introduce learnable prototypes $\{p_m\}_{m=1}^{M}$, with $p_m \in \mathbb{R}^{d_h}$. Given the safety-aware prompt summary, prototype retrieval is performed as

$$q = \phi_q\left(h_{\mathrm{prompt}}^{\mathrm{safe}}\right), \qquad \beta = \mathrm{softmax}\left(P^\top q\right), \qquad h_{\mathrm{proto}} = P\beta, \tag{12}$$

where $P = [p_1, \ldots, p_M] \in \mathbb{R}^{d_h \times M}$. The shared prototype memory provides a global mechanism for reusing recurring transition-related patterns across tasks.

**Final context construction.** Finally, the safety-aware prompt summary and retrieved prototype memory are combined as

$$c_{\text{geo}} = \text{MLP}\left(\left[h_{\text{prompt}}^{\text{safe}}, h_{\text{proto}}\right]\right). \tag{13}$$

The resulting structured context is subsequently fused with complementary task-dependent state features for action prediction.

### 4.3 Cost-Aware Decision Transformer

To generate actions, we combine the structured context with the complementary task-dependent branch. We first encode the complementary component by

$$z_t^{\text{sem}} = \phi_{\text{sem}}(s_t^{\text{sem}}), \tag{14}$$

and fuse it with the structured context:

$$z_t = \text{MLP}([z_t^{\text{sem}}, c_{\text{geo}}]). \tag{15}$$

Following prior work on safe sequence modeling (Liu et al., 2023b; Zhang et al., 2023), we further condition action generation on both return-to-go and cost-to-go. Given a context window of length $H$, the transformer input at timestep $t$ is

$$\mathbf{o}_t = \left(\hat{R}_{t-H:t}, \hat{C}_{t-H:t}, z_{t-H:t}, a_{t-H:t-1}\right), \tag{16}$$

where $\hat{R}_t = \sum_{j=t}^{T} r_j$ and $\hat{C}_t = \sum_{j=t}^{T} c_j$.

We parameterize the policy as a stochastic Gaussian distribution:

$$\pi_\theta(\cdot \mid \mathbf{o}_t) = \mathcal{N}(\mu_\theta(\mathbf{o}_t), \Sigma_\theta(\mathbf{o}_t)), \tag{17}$$

where $\mu_\theta$ and $\Sigma_\theta$ are predicted by a causal Transformer decoder. This design allows GeoDT to leverage shared transition-related structure for cross-task generalization while adapting behavior to different reward and safety requirements.

### 4.4 Training Objective

We train GeoDT end-to-end on offline trajectories using maximum likelihood estimation with entropy regularization (Haarnoja et al., 2018). The policy loss is

$$\mathcal{L}_{\text{pol}} = \mathbb{E}_{\mathbf{o} \sim \mathcal{D}}\left[-\log \pi_\theta(\mathbf{a} \mid \mathbf{o}) - \lambda\, H\big(\pi_\theta(\cdot \mid \mathbf{o})\big)\right], \tag{18}$$

where $\lambda \geq 0$ controls the strength of entropy regularization. The Gaussian policy represents conditional action uncertainty under heterogeneous task objectives and safety budgets, while the entropy term serves only as a mild regularizer rather than the primary safety mechanism. Appendix E.6 separately ablates the stochastic parameterization and entropy regularization. The final training objective combines the sequence modeling loss with the consistency regularizer:

$$\mathcal{L} = \mathcal{L}_{\text{pol}} + \lambda_{\text{geo}}\, \mathcal{L}_{\text{geo}}. \tag{19}$$

During training, trajectory prompts are sampled from the offline dataset to construct context, and no additional environment interaction is required.

---

**Algorithm 1** GeoDT Training

---

**Require:** Task datasets $\mathcal{D} = \bigcup_{i=1}^{N} \mathcal{D}_i$, training iterations $L$, prompt length $K$, learnable prototype memory $\mathcal{P}$

1: **for** $\ell = 1$ to $L$ **do**
2:     Sample a minibatch $\mathcal{B} = \{(\tau_b, \tau_b^\star)\}_{b=1}^{B}$ from the task datasets
3:     **for all** $(\tau_b, \tau_b^\star) \in \mathcal{B}$ **do**
4:         Apply the geometry-inspired decomposition: $(s_b^{\text{geo},\star}, s_b^{\text{sem},\star}) \leftarrow \text{DECOMPOSE}(s_b^\star)$
5:         Encode the routed prompt states: $z_b^{\text{geo},\star} \leftarrow \phi(s_b^{\text{geo},\star})$
6:         Induce within-prompt relations and apply feasibility-aware reweighting using $\hat{C}_b^\star$ to obtain $h_{\text{prompt},b}^{\text{safe}}$
7:         Retrieve and fuse prototype memory:

$$c_{\text{geo},b} \leftarrow \text{MLP}\big(\big[h_{\text{prompt},b}^{\text{safe}}, \text{RETRIEVE}\big(h_{\text{prompt},b}^{\text{safe}}, \mathcal{P}\big)\big]\big)$$

8:         Apply the decomposition to the decision segment and retain its task-dependent component:

$$s_b^{\text{sem}} \leftarrow \text{DECOMPOSESEM}(s_b)$$

9:         Fuse $c_{\text{geo},b}$ with $s_b^{\text{sem}}$ and predict actions using the cost-aware Decision Transformer
10:         Compute $\mathcal{L}_{\text{pol}}^{(b)}$ and $\mathcal{L}_{\text{geo}}^{(b)}$
11:     **end for**
12:     Compute the minibatch objective

$$\mathcal{L} = \frac{1}{B} \sum_{b=1}^{B} \left( \mathcal{L}_{\text{pol}}^{(b)} + \lambda_{\text{geo}} \mathcal{L}_{\text{geo}}^{(b)} \right)$$

13:     Update all model parameters, including $\mathcal{P}$, by gradient descent on $\mathcal{L}$
14: **end for**

---

## 5 Experiment

We evaluate GeoDT on a suite of multi-task safe offline RL benchmarks to assess its ability to (i) remain robust across diverse tasks with heterogeneous semantics, (ii) maintain favorable reward–cost trade-offs under varying safety constraints, and (iii) generalize via trajectory-based context without explicit task identifiers. Implementation details are provided in Appendix C.

**Experiment setup.** Following prior work (Liu et al., 2023b; Achiam et al., 2017; Zhang et al., 2020), we consider four continuous-control safe locomotion tasks: `Run`, `Circle`, `Reach`, and `Gather`. Each task is instantiated with four robot morphologies: `Car`, `Ball`, `Drone`, and `Ant`, resulting in a diverse multi-task setting with varying dynamics, objectives, and safety constraints. All experiments are conducted on the BulletSafetyGym benchmark (Gronauer, 2022). Additional environment details and hyperparameters are provided in Appendix D.

**Offline datasets.** For the `Run` and `Circle` environments, we use the publicly available DSRL dataset (Liu et al., 2023a), which is specifically designed for offline safe RL. For `Reach` and `Gather`, we collect trajectories using a PPO-Lag agent (Ray et al., 2019) under progressively varying cost thresholds to ensure diverse reward–cost trade-offs. All datasets follow the DSRL format, including state, action, reward, cost, and terminal signals, ensuring consistency across tasks.

**Baselines.** We compare GeoDT against a set of strong baselines covering both safe offline RL and multi-task learning. All baselines are adapted to the same multi-task training protocol for fair comparison:

- *CDT* (Liu et al., 2023b): a Decision Transformer variant with cost conditioning.

Table 1: Task-wise performance of *jointly trained multi-task policies*. For each agent family, every method is trained once on all tasks simultaneously, and the resulting single policy is evaluated on each task separately. Higher reward and lower cost are better.

| Method | Car multi-task setting | | | | | | | | Ball multi-task setting | | | | | | | | Average | |
|---|---|---|---|---|---|---|---|---|---|---|---|---|---|---|---|---|---|---|
| | Run | | Circle | | Reach | | Gather | | Run | | Circle | | Reach | | Gather | | Reward ↑ | Cost ↓ |
| | Reward | Cost | Reward | Cost | Reward | Cost | Reward | Cost | Reward | Cost | Reward | Cost | Reward | Cost | Reward | Cost | | |
| GeoDT | **0.95** | **0.77** | **0.73** | **0.61** | **0.70** | **0.67** | 0.41 | 0.43 | 0.74 | 0.74 | **0.75** | **0.63** | **0.87** | **0.79** | 0.48 | 0.63 | **0.704** | **0.659** |
| CDT | 0.94 | 0.79 | 0.68 | 0.56 | 0.69 | 1.27 | **0.43** | **0.86** | 0.73 | 0.80 | 0.76 | 1.21 | 0.83 | 0.73 | 0.45 | 0.55 | 0.686 | 0.836 |
| PDT | 0.93 | 4.42 | 0.68 | 1.96 | **0.63** | **0.65** | 0.40 | 0.63 | **0.88** | **0.97** | 0.69 | 1.25 | 0.84 | 0.94 | 0.45 | 0.50 | 0.685 | 1.414 |
| CPQ | 0.90 | 1.18 | **0.59** | **0.84** | 0.49 | 7.10 | **0.14** | **0.29** | 0.37 | 1.71 | 0.58 | 1.05 | **0.62** | **0.87** | 0.24 | 0.68 | 0.491 | 1.715 |
| BCQL | 0.89 | 1.32 | 0.61 | 3.57 | **0.53** | **0.58** | **0.31** | **0.54** | 0.29 | 0.28 | 0.79 | 1.77 | **0.85** | **0.79** | 0.46 | 0.56 | 0.593 | 1.177 |
| LSPC | **0.80** | **0.64** | 0.42 | 1.87 | **0.28** | **0.29** | 0.22 | 0.55 | 0.91 | 1.10 | 0.62 | 2.38 | 0.44 | 1.29 | **0.25** | **0.45** | 0.492 | 1.069 |
| SoftMod | 0.95 | 2.68 | 0.70 | 4.43 | **0.60** | **0.62** | 0.37 | 0.55 | 1.14 | 1.17 | 0.82 | 2.52 | **0.82** | **0.94** | 0.43 | 0.50 | 0.729 | 1.675 |
| FISOR | **0.22** | **0.33** | **0.28** | **0.26** | 0.51 | 0.69 | 0.40 | 0.15 | **0.02** | **0.03** | 0.37 | 0.01 | 0.74 | 0.97 | **0.51** | **0.45** | 0.379 | 0.360 |
| O3SRL | **0.93** | **0.83** | 0.81 | 2.95 | **0.45** | **0.89** | 0.40 | 0.90 | 1.04 | 1.14 | 0.86 | 2.55 | **0.85** | **0.64** | 0.43 | 0.50 | 0.722 | 1.299 |

*Note:* Results are reported for a *single jointly trained multi-task policy* evaluated separately on each task. **Bold** denotes safe policies, gray denotes unsafe policies, and **blue bold** marks the best safe result.

- *Prompt-DT* (Xu et al., 2022b): a prompt-based Decision Transformer that uses task prompts to support multi-task generalization.

- *LSPC* (Koirala et al., 2024b): a latent-space method balancing reward and cost.

- *CPQ* (Xu et al., 2022a): a Q-learning approach with explicit constraint penalties.

- *BCQ-Lag* (Fujimoto et al., 2019): a Lagrangian extension of BCQ for safe offline RL.

- *SoftMod* (Yang et al., 2020): a modular multi-task architecture, augmented with cost-to-go inputs.

- *FISOR* (Zheng et al., 2024): a diffusion-based method with feasibility guidance.

- *O3SRL* (Chemingui et al., 2025a): an offline safe RL method with online optimization for constraint satisfaction.

**Evaluation metrics.** We evaluate all methods using normalized reward return and normalized cost return, following standard practice in offline RL (Fu et al., 2020; Liu et al., 2023b). The normalized reward is defined as

$$R_{\text{norm}} = \frac{R_\pi - r_{\min}(\mathcal{T})}{r_{\max}(\mathcal{T}) - r_{\min}(\mathcal{T})}, \tag{20}$$

and the normalized cost as

$$C_{\text{norm}} = \frac{C_\pi}{\kappa + \epsilon}, \tag{21}$$

where $R_\pi$ and $C_\pi$ denote the return and cost of policy $\pi$, and $\kappa$ is the task-specific cost threshold.

To further evaluate robustness under increasing task diversity, we additionally report the *Task Scaling Robustness Score (TSRS)* and *Inter-Task Balance Score (ITBS)*, which measure performance degradation and reward–cost balance as the number of tasks increases (see Section 5.2).

## 5.1 Main Multi-Task Performance

Table 1 reports the task-wise performance of a *single jointly trained multi-task policy* for each method. For each agent family, all methods are trained once on the full set of tasks simultaneously, and the resulting policy is then evaluated separately on each task. This setting is substantially more challenging than training independent single-task policies, since the model must resolve cross-task interference while maintaining favorable reward–cost trade-offs under heterogeneous dynamics, objectives, and safety constraints.

Overall, GeoDT achieves the strongest average reward–cost balance among the safe methods considered. In particular, it attains the best average normalized reward (0.704) while also yielding the lowest average normalized cost (0.659) among high-performing methods. This indicates that GeoDT is able to improve

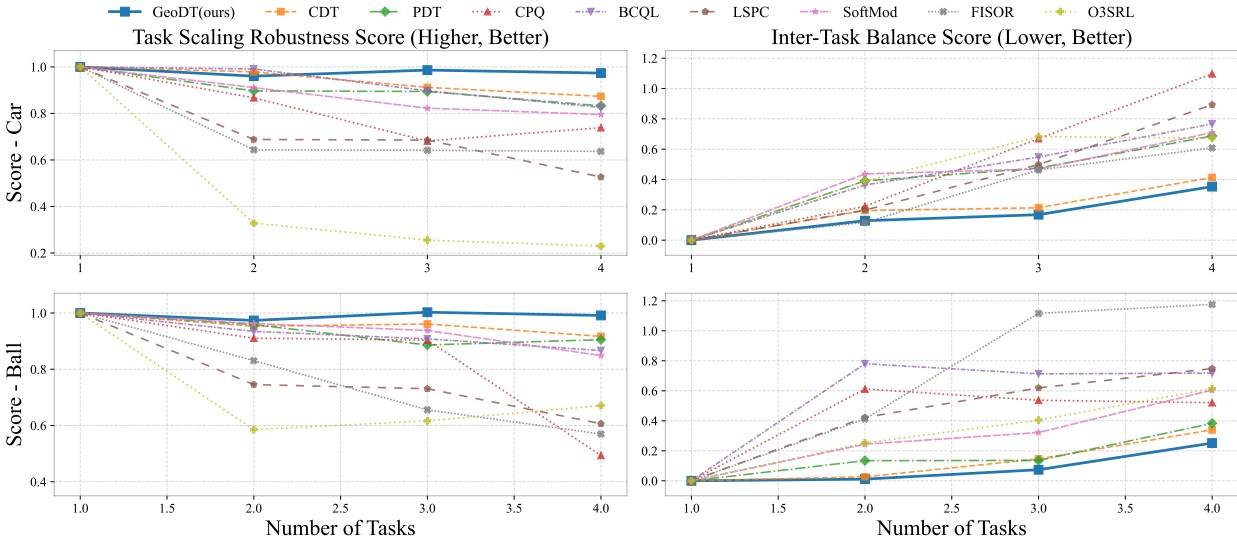

Figure 2: Scaling robustness as the number of jointly trained tasks increases. We report the Task Scaling Robustness Score (TSRS, higher is better) and the Inter-Task Balance Score (ITBS, lower is better). GeoDT consistently maintains stronger robustness and more balanced task-wise behavior than competing methods as task diversity increases.

task performance without sacrificing safety, rather than simply trading lower cost for substantially reduced reward. Compared with CDT and Prompt-DT, which are the strongest sequence-modeling baselines in terms of reward, GeoDT produces more consistent safety behavior and avoids the large cost violations observed in several tasks. Compared with conservative safe RL baselines such as FISOR, GeoDT achieves substantially higher reward while remaining within a competitive safety range.

A more fine-grained view reveals that GeoDT is consistently strong across individual tasks rather than relying on a few outlier wins. It achieves the best safe result on multiple tasks, including Car Run, Car Circle, Car Reach, Ball Circle, and Ball Reach, and remains competitive on the remaining tasks. In contrast, several baselines exhibit a sharper reward–cost imbalance: some methods achieve high reward at the expense of severe safety violations (e.g., O3SRL and SoftMod on several tasks), while others remain safe but become overly conservative and underperform in reward (e.g., FISOR). This pattern suggests that the main difficulty in multi-task safe offline RL is not only maximizing return or minimizing cost in isolation, but balancing the two under cross-task semantic mismatch.

These results are consistent with the central motivation of GeoDT. Through its geometry-inspired decomposition and structured context construction, GeoDT is better able to share transition-related information across tasks without amplifying semantic inconsistency. The resulting benefit is not merely a higher average reward, but a more favorable and stable reward–cost trade-off across heterogeneous multi-task settings. While Table 1 demonstrates the overall effectiveness of GeoDT, average task performance alone does not fully characterize how well a method maintains performance as task diversity increases. We therefore next analyze cross-task robustness more directly using TSRS and ITBS.

## 5.2 Scaling Robustness under Increasing Task Diversity

While Table 1 shows that GeoDT achieves strong overall reward–cost trade-offs, average task performance alone does not fully characterize how well a method maintains performance. In multi-task safe offline RL, a method may achieve strong average performance while still degrading sharply relative to its single-task counterpart, or exhibit large variation across tasks as the number of jointly trained tasks grows. To better characterize robustness under increasing task diversity, we introduce two diagnostic metrics: the **Task Scaling Robustness Score** (TSRS) and the **Inter-Task Balance Score** (ITBS). Our design is motivated

by robustness-oriented statistical measures used to summarize performance degradation and variability in other domains, such as ecological sensing (Xu, 2006) and image quality assessment (Wang et al., 2004).

**Task Scaling Robustness Score (TSRS).** TSRS measures how well a method preserves its overall reward–cost performance relative to the corresponding single-task setting as the number of jointly learned tasks increases. For an $n$-task setting, we define

$$\text{TSRS}(n) = 1 + \frac{1}{\sqrt{n}} \sum_{i=1}^{n} \left[ (1 - \lambda)\delta R_i - \lambda \delta C_i \right], \tag{22}$$

where

$$\delta R_i = \frac{R_i - R_{\text{ref}}}{R_i + R_{\text{ref}} + \varepsilon}, \qquad \delta C_i = \frac{C_i - C_{\text{ref}}}{C_i + C_{\text{ref}} + \varepsilon}. \tag{23}$$

Here, $R_i$ and $C_i$ denote the reward and cost on task $i$ under multi-task training, while $R_{\text{ref}}$ and $C_{\text{ref}}$ are the corresponding single-task reference values. $\varepsilon$ is a small constant for numerical stability, and $\lambda$ controls the reward–cost trade-off; we set $\lambda = 0.5$ in all experiments. A higher TSRS indicates that the method better preserves its single-task performance as task diversity increases. By construction, TSRS = 1 when the multi-task policy exactly matches the corresponding single-task reference in both reward and cost. The score decreases as reward performance degrades relative to the reference or as cost violations increase. The normalization factor $\frac{1}{\sqrt{n}}$ scales the accumulated task-wise deviations, reducing sensitivity to the number of jointly trained tasks and facilitating comparisons across settings with different task counts.

**Inter-Task Balance Score (ITBS).** TSRS evaluates overall performance retention, but does not indicate whether performance is evenly distributed across tasks. To quantify balance, we define

$$\text{ITBS}(n) = \text{CV}(R_1, \dots, R_n) + \lambda \cdot \text{CV}(C_1, \dots, C_n), \tag{24}$$

where CV denotes the coefficient of variation. Lower ITBS indicates more balanced behavior across tasks, with less task-to-task fluctuation in both reward and cost.

Figure 2 reports TSRS and ITBS as the number of jointly trained tasks increases. GeoDT consistently maintains a higher TSRS and a lower ITBS than competing methods, especially in the most challenging four-task setting. Although several baselines remain competitive when only two or three tasks are trained jointly, their performance degrades more sharply as task diversity increases. In contrast, GeoDT maintains performance more effectively, preserving a larger fraction of its single-task performance while maintaining more balanced reward-cost behavior across tasks.

These results are consistent with the design of GeoDT. By routing transition-related information apart from complementary task-dependent information and constructing safety-aware structured context before decision-making, GeoDT reduces cross-task interference and avoids over-specializing to a subset of tasks. As task diversity grows, this yields a more robust and balanced multi-task policy, rather than one that achieves strong performance on only a few tasks at the expense of others.

### 5.3 Zero-Shot Adaptation to Unseen Safety Budgets

An important advantage of return- and cost-conditioned sequence models is their ability to adapt behavior at inference time without retraining. In safe multi-task offline RL, this capability is particularly valuable: a single policy may need to operate under different safety budgets depending on deployment requirements, even when no additional interaction or task-specific fine-tuning is allowed. Following prior work on return- and cost-conditioned decision transformers (Xu et al., 2022b; Liu et al., 2023b), we therefore evaluate whether GeoDT can generalize zero-shot to *unseen target cost budgets* in the multi-task setting.

We compare GeoDT against a multi-task-trained CDT on four tasks for both the `Car` and `Ball` agents. For each task, we sweep 20 target cost budgets at evaluation time. In addition, obstacle layouts and goal positions are randomly regenerated to produce unseen environment configurations, making the evaluation more challenging than simply replaying the offline training distribution. The results are shown in Figure 3.

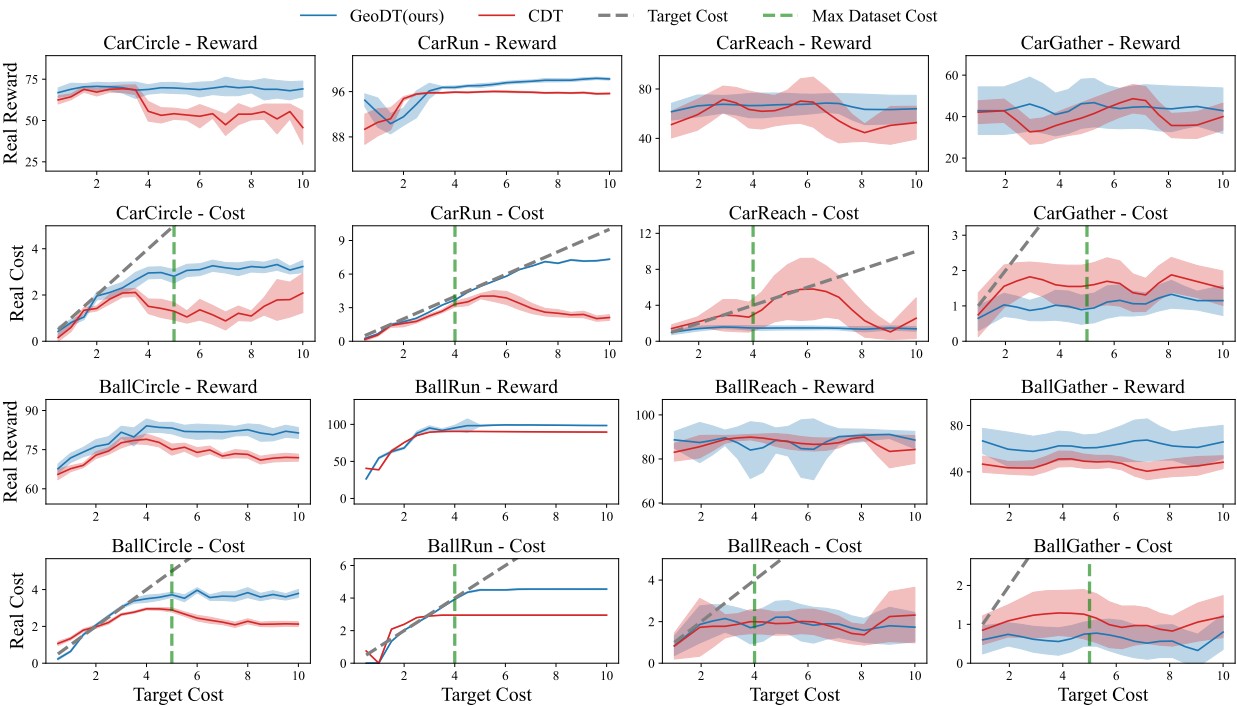

Figure 3: Zero-shot adaptation under varying target cost budgets. Each column corresponds to one task, and the x-axis denotes the target cost. Results are averaged over 20 evaluation episodes and three random seeds. Solid lines show the mean, and shaded regions indicate $\pm 1$ standard deviation. The dashed vertical line marks the maximum offline cost observed in the dataset, and the gray diagonal indicates the ideal relation $y = $ target cost.

Overall, GeoDT achieves a more reliable reward–cost trade-off across a wide range of target budgets. As the target cost increases, GeoDT typically adjusts its behavior smoothly, improving reward when additional cost budget is available while remaining substantially better aligned with the desired cost level. This trend is especially clear in tasks such as `Run` and `Circle`, where reward is strongly coupled with exploratory or aggressive behavior. In these settings, GeoDT adjusts performance more smoothly with the budget, whereas CDT often becomes unstable or violates the intended safety level under tighter constraints.

GeoDT is also more robust when extrapolating beyond the effective training range. Even when target costs approach or exceed the maximum offline cost observed in the dataset, GeoDT maintains comparatively stable behavior and avoids the sharp degradation seen in CDT on several tasks. This suggests that the structured context constructed by GeoDT provides a more transferable representation of safe behavior patterns, allowing the model to adjust to new safety requirements without retraining.

These results complement the multi-task performance and scaling analyses above. Beyond achieving strong average reward–cost trade-offs and robust performance under increasing task diversity, GeoDT also supports zero-shot adaptation to unseen safety budgets, which is essential for practical deployment in safety-critical multi-task settings.

## 5.4 Ablation Study

We perform ablations to evaluate the contribution of the main components in GeoDT, including the geometry-inspired decomposition, prototype memory, and feasibility-aware structure reweighting. Table 2 reports the performance of a single jointly trained multi-task policy for each ablated variant.

Table 2: Ablation of key components in GeoDT. We report average normalized reward and cost over all Car tasks and all Ball tasks, respectively, as well as the overall average. Higher reward and lower cost are better.

| Method | Car Avg | | Ball Avg | | Overall Avg | |
|---|---|---|---|---|---|---|
| | Reward ↑ | Cost ↓ | Reward ↑ | Cost ↓ | Reward ↑ | Cost ↓ |
| GeoDT | **0.697** | **0.620** | **0.710** | **0.698** | **0.704** | **0.659** |
| GeoDT (w/o feasibility weighting) | 0.613 | 0.922 | 0.665 | 0.814 | 0.639 | 0.868 |
| GeoDT (w/o prototype) | 0.610 | 0.783 | 0.660 | 0.691 | 0.635 | 0.737 |
| GeoDT (shared encoding, w/o $\mathcal{L}_{\text{geo}}$) | 0.678 | 0.710 | 0.675 | 0.712 | 0.676 | 0.711 |
| GeoDT (shared encoding, w/ $\mathcal{L}_{\text{geo}}$) | 0.655 | 0.714 | 0.658 | 0.686 | 0.657 | 0.700 |
| GeoDT (decomposition, w/o $\mathcal{L}_{\text{geo}}$) | 0.665 | 0.773 | 0.635 | 0.763 | 0.650 | 0.768 |

*Note:* All variants are trained under the same multi-task protocol. Car Avg and Ball Avg denote averages over the four corresponding tasks for each agent family.

Overall, all major components of GeoDT contribute to the final performance. Removing feasibility-aware reweighting causes the largest increase in average cost, especially on Car tasks, indicating that feasibility-based structure selection is important for maintaining cost compliance. Removing the prototype memory also degrades both reward and cost, showing that the shared prototype memory provides useful structural information beyond prompt-level relations.

The interaction between the geometry-inspired decomposition and the consistency objective is best assessed from the four corresponding factorial configurations. With a shared encoding and without $\mathcal{L}_{\text{geo}}$, the model obtains an overall reward and cost of 0.676/0.711. Applying $\mathcal{L}_{\text{geo}}$ directly to the same undifferentiated representation slightly reduces cost but also lowers reward to 0.657/0.700. Conversely, using the decomposition without $\mathcal{L}_{\text{geo}}$ yields 0.650/0.768. Combining the geometry-inspired decomposition with $\mathcal{L}_{\text{geo}}$ produces 0.704/0.659, outperforming the other three configurations on both metrics. These results indicate that the improvement does not arise from a generic auxiliary prediction objective alone. Rather, $\mathcal{L}_{\text{geo}}$ is most effective when paired with the geometry-inspired decomposition, where it provides a transition-based criterion for shaping the soft routing mask.

Additional experiments on two more agents, as well as more ablation studies and key hyperparameters, are presented in the Appendix E.

## 5.5 What Does the Transition-Related Branch Capture?

A natural concern is whether the geometry-inspired decomposition produces a meaningful transition-related branch, or merely another generic latent representation. We examine this directly with three diagnostics on the four-task Car and Ball settings: branch probing, gate-routing statistics, and a comparison against a fixed manual split.

**Branch probing.** We freeze the trained representations and fit lightweight linear probes on held-out transitions. For dynamics, the probe predicts the next-step agent-centric state variables from $[z_t, a_t]$; for task-dependent information, it predicts the current task-dependent observation variables from $z_t$. We compare the transition-related branch $z_t^{\text{geo}}$, the complementary branch $z_t^{\text{sem}}$, and $z_t^{\text{full}}$, the shared representation from an otherwise matched model trained without decomposition (Table 3). We report the coefficient of determination ($R^2$), for which higher values indicate greater linear predictability. For compactness, the table omits the timestep subscript $t$. The two branches exhibit a clear crossover rather than one representation being uniformly stronger or weaker. On Car, $z_t^{\text{geo}}$ retains nearly all of the dynamics predictability of $z_t^{\text{full}}$ (0.969 vs. 0.981), while its task-dependent predictability drops substantially (0.781 vs. 0.953). Conversely, $z_t^{\text{sem}}$ preserves task-dependent predictability (0.941) but is markedly weaker for dynamics (0.836). Ball shows the same pattern. This crossover provides evidence that the two branches preferentially retain different information, rather than behaving as two undifferentiated generic latents. We do not claim an information-pure partition:

Table 3: Branch probing against the no-decomposition latent ($z_{\text{full}}$). Each entry is the linear-probe $R^2$ for predicting next-step agent-centric dynamics ($R^2_{\text{dyn}}$) or current task-dependent semantics ($R^2_{\text{sem}}$). The crossover between $z_{\text{geo}}$ and $z_{\text{sem}}$ indicates branch selectivity rather than a uniformly stronger latent representation.

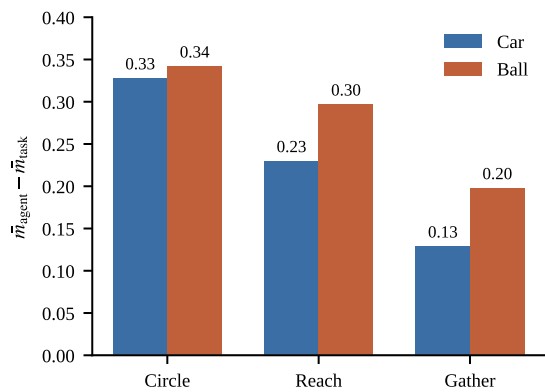

Figure 4: Gate-weight difference between agent-centric and task-dependent dimensions for each task.

|  | Car | | Ball | |
|---|---|---|---|---|
|  | $R^2_{\text{dyn}}$ | $R^2_{\text{sem}}$ | $R^2_{\text{dyn}}$ | $R^2_{\text{sem}}$ |
| $z^{\text{geo}}$ | 0.969 | 0.781 | 0.957 | 0.794 |
| $z^{\text{sem}}$ | 0.836 | 0.941 | 0.861 | 0.907 |
| $z^{\text{full}}$ | 0.981 | 0.953 | 0.971 | 0.956 |

Table 4: Comparison between the learned geometry-inspired decomposition and a fixed, task-agnostic manual split. The manual split assigns the predefined agent-centric dimensions to the transition-related branch and the remaining task-dependent dimensions to the complementary branch. Higher reward and lower cost are better.

| Decomposition | Description | Car Avg | | Ball Avg | | Overall Avg | |
|---|---|---|---|---|---|---|---|
|  |  | $R \uparrow$ | $C \downarrow$ | $R \uparrow$ | $C \downarrow$ | $R \uparrow$ | $C \downarrow$ |
| Learned decomposition | Learned soft mask | 0.697 | **0.620** | **0.710** | **0.698** | **0.704** | **0.659** |
| Manual decomposition | Fixed binary mask | **0.712** | 1.286 | 0.579 | 1.593 | 0.646 | 1.440 |

$z^{\text{geo}}_t$ still retains some task-dependent information, as expected from a soft gate. Our claim is the weaker one of *preferential selectivity*.

**Gate routing.** We further inspect the learned gate directly. Over held-out states, we average the gate value separately over agent-centric and task-dependent observation dimensions. For every task with an explicit task-dependent observation block, agent-centric dimensions receive larger weights toward the transition-related branch. Figure 4 reports $\bar{m}_{\text{agent}} - \bar{m}_{\text{task}}$: the differences are 0.33/0.23/0.13 for Car Circle/Reach/Gather and 0.34/0.30/0.20 for Ball Circle/Reach/Gather. Run is omitted because it has no explicit task-dependent observation dimensions. These routing statistics are consistent with the agent-centric prior while remaining soft and state-dependent.

**Learned versus manual decomposition.** Since the benchmark observations contain identifiable agent-centric and task-dependent components, one might ask whether a fixed manual split could replace the learned gate. We compare the learned decomposition against a fixed, task-agnostic manual split in Table 4. The fixed split is not a reliable substitute: overall, it reduces reward from 0.704 to 0.646 while more than doubling cost from 0.659 to 1.440. On Ball, it sharply degrades both reward ($0.710 \rightarrow 0.579$) and cost ($0.698 \rightarrow 1.593$). On Car, it obtains slightly higher reward (0.712 vs. 0.697), but only by severely sacrificing safety (cost $0.620 \rightarrow 1.286$). Although the learned gate's average preference is consistent with the agent-centric prior, its soft and state-dependent allocation captures finer structure than the hand-crafted binary grouping, which is important for maintaining a reliable reward–cost trade-off.

Together, the probing crossover, input-level gate statistics, and comparison with the manual split provide complementary evidence that the transition-related branch preferentially retains agent-centric, transition-related information rather than functioning as an undifferentiated generic latent representation.

# 6 Conclusion

In this work, we presented **GeoDT** (Geometry-Inspired Decision Transformer), a framework for robust, safe multi-task offline reinforcement learning. GeoDT addresses a central challenge in multi-task offline RL—*semantic inconsistency* across heterogeneous tasks—by decomposing transition-related trajectory structure from task-dependent semantics and using the resulting structured context to guide decision making. In particular, GeoDT combines a learnable soft state decomposition, a self-supervised one-step consistency objective, prototype-augmented context construction, and safety-aware structure reweighting within a cost-conditioned Decision Transformer. Empirically, GeoDT achieves strong reward–cost trade-offs across diverse safe multi-task benchmarks, remains more robust as task diversity increases, and generalizes zero-shot to unseen safety budgets without requiring explicit task identifiers. These results suggest that exploiting shared, transition-related trajectory structure is a promising way to improve robustness and safety in offline multi-task decision making.

**Limitations and future work.** Despite these promising results, several limitations remain. First, GeoDT is still evaluated within task families seen during training; although it generalizes across heterogeneous tasks and unseen safety budgets, transferring to entirely novel task families may still require additional adaptation. Second, our method improves safety through cost-aware structure learning and cost-conditioned decision making, but does not provide hard safety guarantees during deployment. Finally, while the prototype memory is empirically stable, its interpretation remains implicit and could be further studied. Future work will explore stronger forms of safety assurance, including integrating offline safe RL with verifiable control tools such as reachability-based methods (Zheng et al., 2024). Another promising direction is to extend the proposed decomposition to broader cross-domain and cross-embodiment settings, where task semantics vary even more substantially across environments.

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

## A    Discussion on State Structure and Semantic Inconsistency

In the safe locomotion benchmarks considered in this work, the observation vector typically contains two qualitatively different types of information: (1) *agent-centric dynamical variables*, which describe the agent's internal physical state, such as position, velocity, orientation, and actuator status, and (2) *task-dependent environmental signals*, which encode goals, obstacles, boundaries, or other task-specific cues perceived through external sensors.

The first type is generally more stable across tasks for a fixed embodiment. For example, in BulletSafetyGym, the agent-centric part includes variables such as position, velocity, orientation, angular velocity, and motor-related states. These quantities preserve consistent physical semantics across tasks. In contrast, the second type depends strongly on the task definition and observation design, and may vary substantially in both dimensionality and meaning.

For instance, the agent-centric observations are structured as follows:

- `Car`: a 7-dimensional vector including position $(x, y)$, velocity $(v_x, v_y)$, yaw (expressed as sine and cosine), and yaw rate.

- `Ball`: a 7-dimensional vector including position $(x, y)$, velocity $(v_x, v_y)$, and 3-dimensional angular velocity.

- `Drone`: a 17-dimensional vector including position $(x, y, z)$, velocity $(v_x, v_y, v_z)$, a 4-dimensional orientation quaternion, 3-dimensional angular velocity, and motor-related measurements.

- `Ant`: a 33-dimensional vector including position $(x, y, z)$, velocity $(v_x, v_y, v_z)$, a 4-dimensional orientation quaternion, 3-dimensional angular velocity, 16-dimensional motor relative positions, and 4-dimensional leg contact states.

Task-dependent observations are defined as:

- `Run`: no explicit task-specific observations, since the task only requires forward motion under a speed constraint.

- `Circle`: a 1-dimensional value representing the distance to the circle boundary.

- `Reach`: for `Car`, a 26-dimensional vector combining a 2-dimensional goal distance and a 24-dimensional LIDAR scan; for `Ball`, `Drone`, and `Ant`, a 50-dimensional vector combining goal distance with two 24-dimensional LIDAR scans.

- `Gather`: a 32-dimensional vector from two 16-ray LIDAR scans, one detecting goals and the other detecting obstacles.

These examples illustrate why semantic inconsistency naturally arises in multi-task learning. Even when observation vectors have similar dimensionality across tasks, features handled by a shared representation may correspond to different physical quantities or serve different functional roles. For example, one dimension may reflect proximity to a goal in one task but proximity to an obstacle or boundary in another. As a result, naively sharing representations across tasks can introduce ambiguity and conflicting optimization signals.

This observation motivates the design of GeoDT. Rather than manually fixing which dimensions should be shared across tasks, GeoDT learns a soft decomposition that biases one branch toward more stable, transition-related structure while retaining task-dependent semantics in the other, in a data-driven way. The state structure described above therefore serves as intuition for why such a decomposition is beneficial, rather than as a hand-crafted rule used directly by the method.

## B An Illustrative Analysis of Cross-Task Gradient Conflict

### B.1 Simplified Theoretical Analysis

This appendix provides a simplified theoretical analysis to illustrate why semantic inconsistency under shared training can induce cross-task gradient conflict, and why a transition-biased decomposition can help alleviate this effect. The goal is not to fully characterize the optimization dynamics of GeoDT, but to formalize the intuition behind the geometry-inspired decomposition used in Section 4.1.

**Setup.** Consider two tasks $i$ and $j$ trained with a shared representation parameterized by $\theta$. We assume that the input to task $i$ can be decomposed as

$$x_i = z + u_i, \qquad x_j = z + u_j,$$

where $z$ denotes a transition-related component that is shared across tasks, while $u_i$ and $u_j$ denote task-dependent components. Intuitively, $z$ represents the part of the input that provides a more stable basis for cross-task sharing, whereas $u_i$ and $u_j$ represent task-dependent variation that may differ across tasks even when the inputs are processed by a common encoder.

Let the task losses be $\mathcal{L}_i(\theta; x_i)$ and $\mathcal{L}_j(\theta; x_j)$, and let the corresponding gradients on the shared parameters be

$$g_i = \nabla_\theta \mathcal{L}_i(\theta; x_i), \qquad g_j = \nabla_\theta \mathcal{L}_j(\theta; x_j).$$

To quantify cross-task gradient conflict, we consider the expected gradient alignment

$$A_{ij} := \mathbb{E}\big[\langle g_i, g_j \rangle\big].$$

Larger values of $A_{ij}$ indicate better gradient agreement, whereas smaller or negative values indicate stronger conflict.

**Linearized shared training model.**  Suppose the shared encoder directly processes the full input $x_i = z + u_i$. Linearizing the task gradients around the shared component $z$, we write

$$g_i \approx \bar{g}_i + J_i u_i, \qquad g_j \approx \bar{g}_j + J_j u_j,$$

where

$$\bar{g}_i := \nabla_\theta \mathcal{L}_i(\theta; z), \qquad \bar{g}_j := \nabla_\theta \mathcal{L}_j(\theta; z),$$

and $J_i, J_j$ denote the sensitivity of the gradients to task-dependent perturbations.

Expanding the inner product yields

$$\langle g_i, g_j \rangle \approx \langle \bar{g}_i, \bar{g}_j \rangle + \langle \bar{g}_i, J_j u_j \rangle + \langle J_i u_i, \bar{g}_j \rangle + \langle J_i u_i, J_j u_j \rangle. \tag{25}$$

The first term reflects alignment induced by the shared transition-related structure $z$, while the remaining terms arise from task-specific semantic variation.

**Assumptions.**  For simplicity, assume that:

1. $\mathbb{E}[u_i] = \mathbb{E}[u_j] = 0$;

2. the perturbations are independent of the shared component in the linearized regime;

3. $\mathbb{E}[u_i u_j^\top]$ is small when the task-specific semantics differ substantially across tasks.

Under these assumptions, the cross terms involving one perturbation and one shared term vanish in expectation or remain small, so the dominant perturbation effect comes from the last term in equation 25.

**Proposition 1.**  Under the linearized model above, the expected gradient alignment can be decomposed as

$$A_{ij} \approx \mathbb{E}\big[\langle \bar{g}_i, \bar{g}_j \rangle\big] + \mathbb{E}\big[\langle J_i u_i, J_j u_j \rangle\big].$$

Moreover,

$$\big|\mathbb{E}[\langle J_i u_i, J_j u_j \rangle]\big| \leq \|J_i\| \, \|J_j\| \, \mathbb{E}[\|u_i\| \, \|u_j\|].$$

**Interpretation.**  Proposition 1 shows that naive shared encoding mixes two effects: a shared-structure term, which promotes gradient agreement, and a task-specific perturbation term, which scales with both the magnitude of semantic mismatch and the sensitivity of the shared encoder to that mismatch. As task-specific semantics become more heterogeneous, the perturbation term can dominate and reduce gradient alignment, thereby inducing stronger optimization conflict across tasks.

**Proof sketch.**  The decomposition follows by taking expectation in equation 25 and using the assumptions above to suppress the mixed terms. The bound is obtained by Cauchy–Schwarz:

$$|\langle J_i u_i, J_j u_j \rangle| \leq \|J_i u_i\| \, \|J_j u_j\| \leq \|J_i\| \, \|J_j\| \, \|u_i\| \, \|u_j\|,$$

followed by expectation on both sides.

**Effect of the geometry-inspired decomposition.** Now consider a decomposition

$$x_i \mapsto \left(x_i^{\text{geo}}, x_i^{\text{sem}}\right),$$

where the shared branch is encouraged to depend primarily on the transition-related component $z$, while task-dependent variation is preferentially routed to the complementary branch. In the decomposed model, let the shared gradients be

$$\tilde{g}_i \approx \bar{g}_i + \tilde{J}_i u_i, \qquad \tilde{g}_j \approx \bar{g}_j + \tilde{J}_j u_j,$$

where $\tilde{J}_i$ and $\tilde{J}_j$ measure the residual sensitivity of the shared branch to task-dependent perturbations after decomposition.

**Proposition 2.** If the decomposition reduces the sensitivity of the shared branch to task-specific semantic variation, i.e.,

$$\|\tilde{J}_i\| \leq \|J_i\|, \qquad \|\tilde{J}_j\| \leq \|J_j\|,$$

then the perturbation component of the expected gradient alignment satisfies

$$\left|\mathbb{E}[\langle \tilde{J}_i u_i, \tilde{J}_j u_j \rangle]\right| \leq \|\tilde{J}_i\| \, \|\tilde{J}_j\| \, \mathbb{E}[\|u_i\| \, \|u_j\|] \leq \|J_i\| \, \|J_j\| \, \mathbb{E}[\|u_i\| \, \|u_j\|].$$

Therefore, the learned decomposition reduces the magnitude of the task-dependent perturbation term in shared gradients and improves expected gradient alignment relative to naive shared encoding.

**Interpretation.** Proposition 2 formalizes the role of decomposition: it does not need to eliminate task-specific semantic influence entirely. It is enough to reduce the sensitivity of the shared branch to semantic mismatch, thereby shrinking the perturbation term that degrades cross-task gradient agreement.

**Role of the one-step consistency objective.** In GeoDT, the one-step consistency loss further biases the transition-related branch toward action-conditioned, transition-predictive information. In the simplified model above, this can be interpreted as further reducing the sensitivity of the shared branch to task-dependent perturbations, i.e., decreasing $\|\tilde{J}_i\|$ and $\|\tilde{J}_j\|$.

**Corollary 1.** If the one-step consistency objective further reduces the sensitivity of the shared branch to task-dependent perturbations, then it tightens the perturbation bound in Proposition 2 and further improves expected gradient alignment across tasks.

**Discussion.** This analysis suggests that semantic inconsistency hurts multi-task training by injecting task-dependent perturbations into gradients on shared parameters. The learned decomposition helps preserve a cleaner shared optimization signal, while the one-step consistency objective further reduces task-dependent sensitivity in the transition-related branch. Although simplified, this view provides an optimization-based explanation for why a transition-biased decomposition can reduce cross-task interference under heterogeneous shared training.

### B.2 Empirical Gradient Analysis

To complement the simplified analysis above, we further examine optimization behaviour along the shared transition-related pathway during joint training. Specifically, we compare GeoDT with the variant without decomposition on the four-task Car and Ball settings, and compute task-wise gradients on the shared transition-related parameters. This analysis is intended to test whether the learned decomposition alleviates the optimization difficulties caused by semantic inconsistency under naive shared training.

We report three quantities. First, we measure the average pairwise gradient cosine similarity across tasks, where higher values indicate better gradient alignment on the shared branch. Second, we report the fraction of task pairs with negative cosine similarity, where lower values indicate fewer conflicting task updates. Third, we report the average gradient norm along the shared transition-related pathway, which reflects the magnitude of the optimization signal received by the shared parameters. Specifically, gradients are measured on the

Table 5: Empirical gradient analysis along the shared transition-related pathway during joint training. We report the average pairwise gradient cosine similarity, the fraction of task pairs with negative cosine similarity, and the average gradient norm. Higher cosine similarity and gradient norm are better, while a lower negative ratio is better.

| Method | Car | | | Ball | | |
|---|---|---|---|---|---|---|
| | Cosine ↑ | Neg. Ratio ↓ | Grad Norm ↑ | Cosine ↑ | Neg. Ratio ↓ | Grad Norm ↑ |
| GeoDT | 0.0420 | 0.4113 | 5.386 | 0.0308 | 0.3300 | 2.526 |
| w/o decomposition | -0.0300 | 0.7500 | 0.079 | 0.0013 | 0.5000 | 0.145 |

decomposition, transition-related encoder, relational structure, and prototype-context parameters, and the reported values are averaged over the last 10 logging points and 3 seeds.

As shown in Table 5, GeoDT achieves higher gradient cosine similarity, a lower fraction of conflicting task pairs, and substantially larger shared-branch gradient norms than the variant without decomposition on both Car and Ball. In particular, the negative-pair ratio is reduced by 45.2% on Car and 34.0% on Ball. These empirical observations are consistent with the simplified analysis above: semantic inconsistency under naive shared training can induce more conflicting task updates and weaken the effective optimization signal along the shared pathway, whereas the learned decomposition helps preserve more stable and informative shared updates.

## C  Implementation Details

### C.1  Data Collection Algorithm

We construct our dataset using the PPO-Lag algorithm (Ray et al., 2019), an extension of the PPO method to constrained reinforcement learning.

The learning problem is formulated as a Constrained Markov Decision Process (CMDP), where the objective is to maximize expected returns while satisfying predefined safety constraints. To incorporate the constraint, a non-negative Lagrange multiplier $\lambda$ is introduced, yielding the following unconstrained Lagrangian objective:

$$\mathcal{L}(\theta, \lambda) = J_R(\theta) - \lambda \left( J_C(\theta) - d \right). \tag{26}$$

where $d$ is the cost budget. PPO-Lag performs on-policy updates by optimizing a clipped surrogate objective with a Lagrangian-modified advantage function:

$$L^{\text{PPO-Lag}}(\theta) = \mathbb{E}_t \left[ \min \left( r_t(\theta) \, \hat{A}_t^{\text{Lag}}, \quad \text{clip}\left( r_t(\theta), 1 - \epsilon, 1 + \epsilon \right) \hat{A}_t^{\text{Lag}} \right) \right], \tag{27}$$

$$\hat{A}_t^{\text{Lag}} = \hat{A}_{R,t} \, - \, \lambda \, \hat{A}_{C,t}.$$

Here, $\hat{A}_{R,t}$ and $\hat{A}_{C,t}$ denote empirical estimates of the advantage functions with respect to the reward and cost signals, respectively. These quantify the relative benefit of an action compared to the policy's average behavior under each signal, and are typically estimated using a value baseline or generalized advantage estimation (GAE) (Schulman et al., 2015).

The Lagrange multiplier is updated via projected gradient ascent:

$$\lambda \leftarrow \left[ \lambda + \alpha_\lambda \left( J_C(\theta) - d \right) \right]_+ \tag{28}$$

where $[\cdot]_+$ denotes projection onto the non-negative orthant. The code implementation is adapted from the FSRL (Liu et al., 2024) framework.

## C.2 Dataset Specifications

The statistics of the dataset for each task are summarized in Table 6.

Table 6: Dataset statistics across domains and tasks.

| Domain | Task | Trajectories | Average Reward | Average Cost |
|--------|------|-------------|----------------|--------------|
| Car | Circle | 1450 | $369.9 \pm 141.9$ | $49.8 \pm 30.5$ |
| | Run | 651 | $508.3 \pm 66.6$ | $14.2 \pm 11.6$ |
| | Reach | 1404 | $57.1 \pm 17.5$ | $9.0 \pm 8.7$ |
| | Gather | 1260 | $30.3 \pm 9.4$ | $1.6 \pm 1.2$ |
| Ball | Circle | 886 | $591.6 \pm 220.9$ | $43.1 \pm 22.2$ |
| | Run | 940 | $618.6 \pm 270.1$ | $48.7 \pm 23.1$ |
| | Reach | 1456 | $381.1 \pm 48.8$ | $16.2 \pm 11.6$ |
| | Gather | 1253 | $29.6 \pm 8.4$ | $1.1 \pm 1.0$ |
| Drone | Circle | 1923 | $768.8 \pm 149.0$ | $53.8 \pm 32.9$ |
| | Run | 1990 | $396.4 \pm 101.3$ | $50.5 \pm 43.2$ |
| | Reach | 936 | $119.3 \pm 93.9$ | $15.6 \pm 11.7$ |
| | Gather | 751 | $12.6 \pm 5.0$ | $0.5 \pm 0.7$ |
| Ant | Circle | 5728 | $239.0 \pm 108.3$ | $85.3 \pm 54.9$ |
| | Run | 1816 | $601.6 \pm 170.6$ | $68.1 \pm 47.1$ |
| | Reach | 847 | $10.4 \pm 12.8$ | $9.3 \pm 12.1$ |
| | Gather | 656 | $12.7 \pm 4.4$ | $0.4 \pm 0.6$ |

*Note:* Values are reported as mean $\pm$ standard deviation.

The offline datasets exhibit substantial return variability under the imposed collection cost limits. For instance, the standard deviation of episodic returns reaches $\pm 141.9$ for Car–Circle, $\pm 270.1$ for Ball–Run, and $\pm 149.0$ for Drone–Circle, indicating that policy performance fluctuates widely as it strives to satisfy the cost constraint. By contrast, the simpler Reach and Gather tasks yield markedly lower cost dispersion—approximately $\pm 8$–12 for Reach and only $\pm 0.6$–1.2 for Gather, reflecting their reduced complexity and hence diminished sensitivity to the cost limit, which results in more stable average performance.

## C.3 Training Details

We provide additional implementation details for GeoDT in this appendix. The implementation and the associated datasets are available at `https://anonymous.4open.science/r/GeoDT-A09C`. For each dataset, 10% of the trajectories are randomly reserved for evaluation, while the remaining 90% are used for training. During both training and evaluation, prompts are sampled from the corresponding split. All experiments are conducted on eight NVIDIA RTX 6000 Ada Generation GPUs, each with 48 GB of memory. Unless otherwise stated, all reported results are averaged over 3 random seeds, and each evaluation is performed over 20 episodes per seed.

Algorithms 1 and 2 summarize the training and inference procedures of GeoDT, respectively. During training, the routing modules, structured context encoder, prototype memory, and policy are optimized jointly. During inference, the prompt remains fixed throughout an episode; therefore, its structured context $c_{\text{geo}}$ is computed once and cached. At each timestep, GeoDT only routes the current state, fuses its complementary task-dependent features with the cached context, and performs causal action prediction.

---

**Algorithm 2** GeoDT Inference

---

**Require:** Trained GeoDT, fixed prompt $\tau^\star$, context length $H$, initial target return $\hat{R}_1$, initial cost budget $\hat{C}_1$, episode horizon $T$

1: Compute and cache the structured prompt context:

$$c_{\text{geo}} \leftarrow \text{BUILDCONTEXT}(\tau^\star)$$

2: Initialize trajectory history $\mathcal{H}_0 \leftarrow \varnothing$
3: **for** $t = 1$ to $T$ **do**
4:     Observe the current state $s_t$
5:     Route and encode the complementary task-dependent component:

$$s_t^{\text{sem}} \leftarrow \text{DECOMPOSESEM}(s_t), \qquad z_t^{\text{sem}} \leftarrow \phi_{\text{sem}}(s_t^{\text{sem}})$$

6:     Fuse the current features with the cached prompt context:

$$z_t \leftarrow \text{MLP}\left([z_t^{\text{sem}}, c_{\text{geo}}]\right)$$

7:     Construct $\mathbf{o}_t$ from the most recent $H$ steps of returns-to-go, costs-to-go, fused states, and previous actions
8:     Predict the action distribution $\pi_\theta(\cdot \mid \mathbf{o}_t)$
9:     Select the mean action $a_t \leftarrow \mu_\theta(\mathbf{o}_t)$ and execute it
10:     Observe reward $r_t$, cost $c_t$, and next state $s_{t+1}$
11:     Update the trajectory history $\mathcal{H}_t \leftarrow \mathcal{H}_{t-1} \cup \{s_t, a_t, r_t, c_t\}$
12:     Update the conditioning targets:

$$\hat{R}_{t+1} \leftarrow \hat{R}_t - r_t, \qquad \hat{C}_{t+1} \leftarrow \hat{C}_t - c_t$$

13: **end for**

---

When the optional prototype balancing regularizer is enabled in the appendix study, we further add $\lambda_{\text{bal}}\mathcal{L}_{\text{bal}}$ to the total objective.

## D   Experiment Setting and Hyperparameters

### D.1   Environment Task Description

We use BulletSafetyGym (Gronauer, 2022) environments for training and evaluation. In the `Circle` tasks, the agent is expected to move on a circle in a clockwise direction and is rewarded based on its velocity and proximity to the boundary of the circle. The reward and cost are defined as:

$$r\left(s_{\boldsymbol{t}}\right) = 0.1 \times \frac{-y_t v_x + x_t v_y}{1 + \left|\|x_{\boldsymbol{t}}\|_2 - r\right|} + 0.01 \times r_{\text{agent}}\left(s_{\boldsymbol{t}}\right)$$
$$c\left(s_{\boldsymbol{t}}\right) = \mathbf{1}\left(|x_{\boldsymbol{t}}| > x_{\text{lim}}\right)$$

(29)

where $r$ is the radius of the circle, $x_{\text{lim}}$ is the safety region, and $r_{\text{agent}}(\cdot)$ represents additional rewards based on the agent state, such as electricity costs, speed costs, joint cost, etc.

In the `Run` tasks, the agent is expected to run through an avenue between two safety boundaries. The reward and cost are defined as:

$$r\left(s_{\boldsymbol{t}}\right) = \left(\phi_t - \phi_{t-1}\right) + r_{\text{agent}}\left(s_{\boldsymbol{t}}\right)$$
$$c\left(s_{\boldsymbol{t}}\right) = \min(1, \mathbf{1}\left(|y_t| > y_{\text{lim}}\right) + \mathbf{1}\left(\|\boldsymbol{v_t}\|_2 > v_{\text{lim}}\right))$$

(30)

where $\phi_t = -60 \times \|(x_t, y_t) - (g_x, 0)\|_2$ is the task potential to a fictitious target position $(g_x, 0)$, $y_{\text{lim}}$ is the safety region, and $v_{\text{lim}}$ is the speed limit.

In the `Reach` tasks, the agent is supposed to move towards a series of goals. Obstacles are placed to hinder the agent from trivial solutions. The agent is rewarded for moving closer to the goal and entering the goal zone. The reward and cost are defined as:

$$
\begin{aligned}
r\left(s_{\boldsymbol{t}}\right) &= \left(\phi_t - \phi_{t-1}\right) + r_{\text{agent}}\left(s_{\boldsymbol{t}}\right) \\
c\left(s_{\boldsymbol{t}}\right) &= \min(1, N_t + \mathbf{1}\left(\boldsymbol{z_t} > z_{\text{lim}}\right))
\end{aligned}
\tag{31}
$$

where $\phi_t = -\left\|(x_t, y_t) - (g_x, g_y)\right\|_2$ is the task potential to the current goal $(g_x, g_y)$, $N_t$ is the number of collisions and $z_{\text{lim}}$ is the height limit.

In the `Gather` tasks, the agent is expected to navigate and collect as many green apples as possible while avoiding red bombs. In contrast to the other tasks, agents in the gather tasks receive only sparse rewards when reaching apples. Costs are also sparse and are received when touching bombs. Let the agent's 2D position be $\mathbf{p}_t = (x_t, y_t)$ and let $\mathcal{O}_{\text{apple}}$ denote the set of all apple-type obstacles. Let $\delta_t(o) = \mathbb{I}\left[\text{visible}_t(o) \wedge d_{o,t} < d_{\text{det}}\right]$. The reward is defined as:

$$
r(s_t) = \begin{cases} R_{\text{dead}}, & \text{if the agent is dead at } t, \\ R_{\text{apple}} \sum_{o \in \mathcal{O}_{\text{apple}}} \delta_t(o), & \text{otherwise} \end{cases}
\tag{32}
$$

where $d_{o,t} = \left\|\mathbf{p}_{o,t} - \mathbf{p}_t\right\|$ is the agent distance to apple. The cost is defined as:

$$
c\left(s_{\boldsymbol{t}}\right) = \min(1, N_t \cdot B + \mathbf{1}\left(\boldsymbol{z_t} > z_{\text{lim}}\right))
\tag{33}
$$

where $N_t$ is the number of bombs touched at time $t$ and $B$ the per-bomb penalty.

### D.2 Hyperparameters

For all baseline methods, we adopt their default hyperparameter configurations. To ensure a fair comparison across all methods, we set the rollout length for each task to match the maximum number of interaction steps used by GeoDT and the baselines. The cost threshold for the baselines is set to 10 for all tasks, except for the `Gather` task, where it is set to 1. The key hyperparameters used for the Causal Transformer-based models (GeoDT, CDT and PDT) are provided in Table 7. The key hyperparameters used for other baselines are shown in Table 8. We apply the same setting for multi-task and single-task cases.

Table 7: Causal Transformer–based model configuration parameters

| Parameter | GeoDT | CDT | PDT |
|---|---|---|---|
| *Common Settings:* | | | |
| Embedding dimension | | 128 | |
| Batch size | | 512 | |
| Dropout | | 0.1 | |
| *Architecture:* | | | |
| Number of layers | 3 | 3 | 3 |
| Number of attention heads | 8 | 8 | 1 |
| Context length | 20 | 10 | 20 |
| Prompt length | 20 | – | 20 |
| *Optimization:* | | | |
| Learning rate ($\times 10^{-4}$) | 1.0 | 1.0 | 1.0 |
| Grad norm clip | 0.25 | 0.25 | 0.50 |

Table 8: Other baseline model configuration parameters

| Parameter | CPQ | BCQ-L | LSPC | SoftMod | FISOR | O3SRL |
|---|---|---|---|---|---|---|
| *Common Settings:* | | | | | | |
| Training steps | | | | $1 \times 10^6$ | | |
| Batch size | | | | 512 | | |
| Dropout | | | | 0.1 | | |
| *Algorithm-Specific Settings:* | | | | | | |
| Hidden layer size | 256 | 256 | 256 | 64 | 256 | 256 |
| Soft update rate ($\tau$) | 0.005 | 0.005 | 0.005 | 0.005 | 0.001 | 0.005 |
| *Learning Rates ($\times 10^{-3}$):* | | | | | | |
| Actor learning rate | 1.0 | 1.0 | 0.3 | 0.1 | 0.3 | 0.3 |
| Critic learning rate | 1.0 | 1.0 | 0.3 | 0.1 | 0.3 | 0.3 |
| Activation function | ReLU | ReLU | ReLU | ReLU | *Mish* | ReLU |

# E  Additional Experiments Results

## E.1  Experiment Results on Agent `Drone` and `Ant`

We further evaluate the effectiveness of GeoDT on the more complex `Drone` agent, which features higher-dimensional observations and actions across tasks. We conduct multi-task experiments with the number of tasks to be learned ranging from 2 to 4. Table 9 presents the performance of various methods on the full set of four tasks: `Run`, `Circle`, `Reach`, and `Gather`.

Table 9: Performance comparison with the `Drone` agent across tasks.

| Methods | Drone-Run | | Drone-Circle | | Drone-Reach | | Drone-Gather | | Average | |
|---|---|---|---|---|---|---|---|---|---|---|
| | Reward | Cost | Reward | Cost | Reward | Cost | Reward | Cost | Reward | Cost |
| GeoDT | **0.67** | **0.89** | **0.75** | **0.86** | **0.33** | 0.92 | **0.16** | 0.15 | **0.477** | **0.704** |
| CDT | **0.65** | 0.38 | **0.72** | 0.79 | 0.40 | 1.29 | **0.13** | 0.30 | 0.474 | 0.690 |
| PDT | **0.43** | 0.45 | **0.38** | 0.72 | **0.13** | 0.79 | **0.12** | 0.76 | 0.262 | 0.678 |
| CPQ | 0.85 | 1.42 | 0.51 | 1.78 | 0.21 | 1.32 | **0.12** | 0.23 | 0.419 | 1.188 |
| BCQL | 0.79 | 1.20 | 0.83 | 3.41 | **0.35** | **0.85** | 0.09 | 0.33 | 0.517 | 1.449 |
| LSPC | **0.61** | **0.75** | 0.85 | 1.834 | 0.30 | 1.43 | **0.21** | 0.15 | 0.493 | 1.041 |
| SoftMod | 0.91 | 1.51 | 0.69 | 4.93 | **0.28** | **0.90** | **0.13** | 0.15 | 0.502 | 1.872 |
| FISOR | **0.02** | **0.91** | **0.37** | 0.07 | **0.32** | 0.78 | **0.14** | 0.08 | **0.211** | 0.459 |
| O3SRL | **0.47** | **0.74** | 0.82 | 1.83 | 0.43 | 1.15 | **0.15** | 0.50 | 0.469 | 1.054 |

Compared with agent `Drone`, `Ant` is even more complex, which has the highest state dimension over all the SafetyBulletGym environments. Following the same evaluation protocol as in previous experiments, we compare the performance of GeoDT and other baselines on the full set of four tasks: `Run`, `Circle`, `Reach`, and `Gather`, the results of which are shown in Table 10.

For both agents, despite the high-dimensional state space, GeoDT demonstrates strong multi-task performance across all baselines, maintaining safety satisfaction in all tasks, highlighting the effectiveness of the GeoDT framework.

## E.2  Sensitivity to the Number of Prototypes

We further study sensitivity to the number of prototypes $M$ on two representative agents, `Car` and `Ant`. As shown in Figure 5, GeoDT remains relatively stable across $M \in \{4, 8, 16\}$, indicating that the prototype memory is not overly sensitive to the exact choice of $M$. For `Car`, the average reward varies only slightly

Table 10: Performance comparison with the `Ant` agent across tasks.

| Methods | Ant-Run | | Ant-Circle | | Ant-Reach | | Ant-Gather | | Average | |
|---|---|---|---|---|---|---|---|---|---|---|
| | Reward | Cost | Reward | Cost | Reward | Cost | Reward | Cost | Reward | Cost |
| GeoDT | **0.86** | **0.25** | **0.54** | **0.87** | **0.27** | **0.31** | **0.15** | **0.05** | **0.455** | **0.370** |
| CDT | **0.91** | **0.61** | 0.81 | 1.63 | 0.07 | 1.79 | **0.10** | **0.05** | 0.472 | 1.020 |
| PDT | **0.23** | **0.03** | **0.10** | **0.31** | 0.05 | 1.13 | **0.02** | **0.03** | **0.100** | **0.375** |
| CPQ | **0.10** | **0.00** | **0.02** | **0.00** | **-0.10** | **0.95** | **-0.28** | **0.21** | **-0.066** | **0.245** |
| BCQL | 1.26 | 7.74 | 0.68 | 3.12 | **0.07** | **0.82** | **-0.10** | **0.03** | 0.479 | 2.924 |
| LSPC | 0.86 | 2.19 | 1.02 | 3.11 | **0.22** | **0.26** | **0.20** | **0.00** | 0.573 | **1.388** |
| SoftMod | 0.71 | 4.18 | 0.87 | 3.61 | **0.18** | **0.46** | **0.10** | **0.05** | 0.463 | 2.073 |
| FISOR | **0.17** | **0.01** | **0.22** | **0.00** | **0.17** | **0.58** | **0.15** | **0.03** | **0.178** | **0.153** |
| O3SRL | **0.50** | **0.19** | 0.90 | 4.32 | **0.36** | **0.23** | **0.10** | **0.00** | 0.464 | 1.183 |

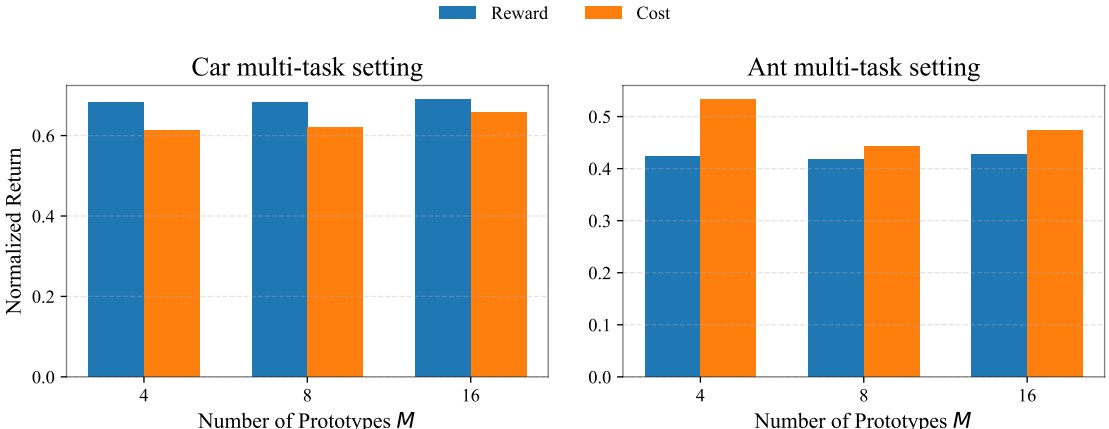

Figure 5: Sensitivity to the number of prototypes $M$ on representative agents. We report the average normalized reward and cost over all tasks for the Car and Ant multi-task settings. GeoDT remains stable across a moderate range of $M$, indicating that the prototype memory is not overly sensitive to this hyperparameter.

across different prototype counts, while the cost remains within a relatively narrow range, suggesting that the shared prototype memory remains effective once moderate capacity is provided. A similar trend is observed for `Ant`, where different values of $M$ lead to comparable reward–cost trade-offs without dramatic degradation. Overall, these results suggest that the gains of GeoDT do not rely on careful tuning of the prototype count; rather, a moderate number of prototypes is sufficient to represent useful recurring transition-related patterns. We therefore use $M = 8$ in the main experiments as a balanced default choice.

### E.3  How Does the Prompt Length Affect the Performance?

Our method uses trajectory prompts both to identify the current task and to guide the agent's behaviour. In this section, we evaluate how prompt length influences performance by comparing multi-task results for the `Car` and `Ball` agents with prompt lengths of 10, 20, and 40. The results are summarized in Table 11 and Table 12.

At the beginning of each episode, the trajectory prompt is randomly selected from the offline dataset, without bias toward high-reward or low-cost segments. Although one might expect that selecting such prompts could enhance performance, we did not observe clear improvements in preliminary experiments. However, it is shown that the prompt length does make a difference, and its effectiveness depends critically on the quality and diversity of the offline datasets and must be interpreted on a per-task basis—it is by no means "the longer,

Table 11: Prompt length effect on the `Car` agent across tasks.

| Prompt Len | Car-Run | | Car-Circle | | Car-Reach | | Car-Gather | | Average | |
|---|---|---|---|---|---|---|---|---|---|---|
| | Reward | Cost | Reward | Cost | Reward | Cost | Reward | Cost | Reward | Cost |
| 10 | 0.951 | 1.345 | 0.717 | 0.813 | 0.666 | 0.638 | 0.358 | 0.600 | 0.673 | 0.849 |
| 20 | 0.947 | 0.770 | 0.726 | 0.614 | 0.701 | 0.673 | 0.413 | 0.420 | 0.697 | 0.620 |
| 40 | 0.943 | 0.645 | 0.731 | 0.608 | 0.650 | 0.630 | 0.367 | 0.625 | 0.673 | 0.627 |

Table 12: Prompt length effect on the `Ball` agent across tasks.

| Prompt Len | Ball-Run | | Ball-Circle | | Ball-Reach | | Ball-Gather | | Average | |
|---|---|---|---|---|---|---|---|---|---|---|
| | Reward | Cost | Reward | Cost | Reward | Cost | Reward | Cost | Reward | Cost |
| 10 | 0.804 | 1.128 | 0.737 | 0.650 | 0.895 | 0.890 | 0.525 | 0.400 | 0.740 | 0.767 |
| 20 | 0.740 | 0.743 | 0.745 | 0.627 | 0.874 | 0.785 | 0.480 | 0.630 | 0.710 | 0.696 |
| 40 | 0.753 | 0.903 | 0.762 | 0.635 | 0.869 | 0.658 | 0.492 | 0.575 | 0.720 | 0.693 |

the better." For dynamically simple tasks (e.g., `Run` and `Circle`), the dataset typically contains high-quality, highly repetitive trajectories; longer prompts can thus leverage safer and more efficient behaviour examples, enabling the Transformer to better capture safety patterns and achieve lower costs or higher rewards. Indeed, as prompt length increases, the safety of `Ball-Run`, `Car-Run`, and `Car-Circle` improves, while the reward for `Ball-Circle` and `Car-Circle` rises markedly. By contrast, for dynamically complex tasks (e.g., `Reach` and `Gather`), where obstacle layouts and goal positions vary randomly, and data noise is significant, overly long contexts fail to mitigate distributional shift and instead introduce irrelevant trajectory fragments. The resulting noise in task identification or policy generation confuses the model and degrades performance. Consequently, shorter or medium-length prompts (10 or 20) prove more robust for these tasks.

In summary, prompt length should be tailored to task complexity to balance information richness against noise interference. Simple tasks benefit from longer prompts that provide more positive examples, whereas complex tasks require restraint to avoid diluting critical context. In our experiments, a prompt length of 20 consistently achieved the lowest cost across most tasks, suggesting it strikes an optimal trade-off between capturing safety patterns and minimizing redundant interference.

### E.4 Optional Prototype Balancing Regularization

In GeoDT, the prototype memory is retrieved through a soft assignment mechanism, which may, in principle, become overly concentrated on a small subset of prototypes. To examine whether a lightweight balancing regularizer can improve memory utilization, we additionally consider an optional prototype balancing loss.

Let $\beta \in \mathbb{R}^{B \times K}$ denote the prototype retrieval weights for a batch, where $K$ is the number of prototypes. We compute the average prototype usage

$$\bar{\beta}_k = \frac{1}{B} \sum_{b=1}^{B} \beta_k^{(b)}, \tag{34}$$

and define a balancing regularizer as

$$\mathcal{L}_{\text{bal}} = \sum_{k=1}^{K} \bar{\beta}_k \log \bar{\beta}_k, \tag{35}$$

which encourages a less degenerate usage distribution over the prototype memory. The overall objective becomes

$$\mathcal{L} = \mathcal{L}_{\text{DT}} + \lambda_{\text{geo}} \mathcal{L}_{\text{geo}} + \lambda_{\text{bal}} \mathcal{L}_{\text{bal}}. \tag{36}$$

We evaluate this variant with $K = 8$, which is the default setting used in the main experiments. As shown in Table 13, adding prototype balancing leads to only minor changes in reward but tends to increase the

Table 13: Effect of optional prototype balancing regularization with $K = 8$. While balancing slightly changes the average reward, it tends to increase realized cost, leading to no clear improvement in the overall reward-cost trade-off.

| Method | Car Avg | | Ant Avg | | Overall Avg | |
|---|---|---|---|---|---|---|
| | Reward | Cost | Reward | Cost | Reward | Cost |
| GeoDT ($K = 8$) | 0.697 | 0.620 | 0.418 | 0.443 | 0.558 | 0.532 |
| GeoDT + balance ($K = 8$) | 0.695 | 0.718 | 0.428 | 0.52 | 0.562 | 0.619 |

realized cost. In other words, encouraging a more even prototype usage distribution does not improve the overall reward-cost trade-off in our setting, even though it can slightly alter the retrieval behavior of the memory module.

This observation is consistent with the design of GeoDT. The prototype memory is intended to provide a compact set of reusable transition-related anchors, but we do not require these prototypes to be maximally distinct or uniformly utilized. Some overlap in prototype usage is natural, since different tasks may still share partially similar transition-related structure. We therefore treat prototype balancing as an optional regularizer rather than a core component of the method, and keep the default GeoDT formulation as simple as possible in the main paper. Indeed, GeoDT does not rely on enforcing prototypes to be maximally dissimilar; rather, it only requires them to provide a useful and reusable set of structural reference patterns.

### E.5 How Does the Safety Coefficient Affect TSRS and ITBS?

In section 5.2, we initially evaluated all methods using a fixed safe coefficient $\lambda = 0.5$ to balance reward and cost in a principled way. To better understand how this balance influences comparative performance, we conducted a sensitivity analysis by sweeping $\lambda \in \{0.0, 0.25, 0.5, 0.75, 1.0\}$ and computing TSRS and ITBS for each method at $n = 4$ in both the Car and Ball environments.

As shown in Figure 6, the results demonstrate that our method yields the smoothest TSRS curves and the smallest ITBS fluctuations across all $\lambda$ values and in both environments, consistently tracking close to the ideal reference. This remarkable insensitivity to $\lambda$ confirms that GeoDT's internal reward–cost balancing mechanism and safety control remain robust even under extreme weightings. Moreover, we find that $\lambda \approx 0.5$ offers an optimal compromise between model separation and ranking stability: at lower $\lambda$, TSRS gradually decreases (improving reward alignment) while ITBS increases (focusing greater on cost variability), yet GeoDT maintains its characteristic flat profile around this point.

For applications prioritizing maximal differentiation in the `Car` environment, raising $\lambda$ toward the 0.75–1.0 range is effective; conversely, in the Ball environment, selecting $\lambda \in [0.25, 0.5]$ better suppresses inter-task volatility. These findings not only justify GeoDT's default choice of $\lambda = 0.5$ but also provide practical guidance for dynamically tuning $\lambda$ in other methods to balance discriminative power against performance consistency.

### E.6 Effect of the Stochastic Policy and Entropy Regularizer

We separate two design choices that are easily conflated: the stochastic Gaussian policy parameterization, and the mild entropy bonus used during training. Table 14 ablates them separately. The *w/o entropy term* variant keeps the Gaussian policy but sets $\lambda = 0$; the *deterministic* variant replaces the Gaussian parameterization with deterministic action regression and is retrained accordingly, with all other components unchanged.

The results suggest that the stochastic parameterization contributes more strongly to the observed safety trade-off than the entropy bonus. Following CDT Liu et al. (2023b)/ODT Zheng et al. (2022)-style decision transformers, the Gaussian policy lets the model represent conditional action uncertainty, relevant when similar states or prompts admit different valid actions across task objectives and cost budgets, while the likelihood objective keeps probability mass anchored to observed actions. Replacing it with a deterministic

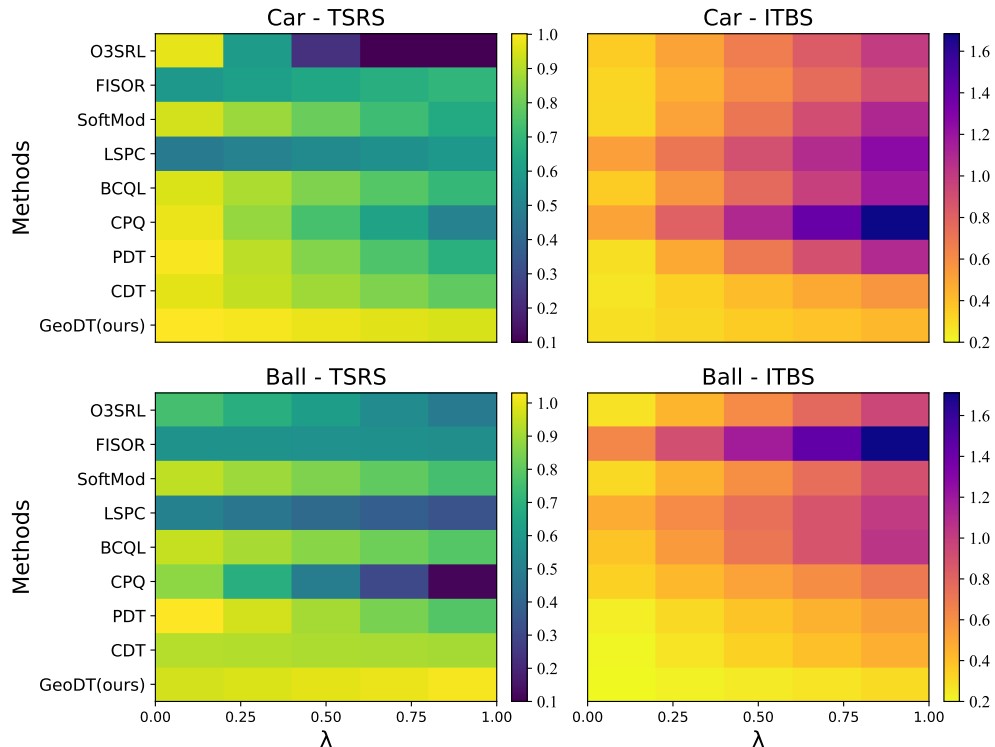

Figure 6: Effect of the safety coefficient sweep on TSRS and ITBS. As the safety coefficient $\lambda$ increases, TSRS tends to decrease while ITBS increases. A moderate value, such as $\lambda = 0.5$, offers a reasonable trade-off between model separation and ranking stability. Notably, GeoDT consistently achieves the highest TSRS and lowest ITBS across all settings, demonstrating its robustness to varying safety weights and, more importantly, its balanced response to reward and cost trade-offs.

Table 14: Effect of disabling the entropy bonus and the stochastic policy. Higher normalized reward $R_{\mathrm{norm}}$ and lower normalized cost $C_{\mathrm{norm}}$ are better.

| Method | Car Avg | | Ball Avg | | Overall Avg | |
|---|---|---|---|---|---|---|
| | Reward↑ | Cost↓ | Reward↑ | Cost↓ | Reward↑ | Cost↓ |
| GeoDT | 0.697 | 0.620 | 0.710 | 0.698 | 0.704 | 0.659 |
| GeoDT (w/o entropy term) | 0.654 | 0.656 | 0.684 | 0.731 | 0.669 | 0.694 |
| GeoDT (deterministic) | 0.821 | 1.347 | 0.759 | 1.892 | 0.790 | 1.620 |

policy sharply degrades the trade-off, raising overall normalized cost from 0.659 to 1.620 and pushing the policy well outside the safe region despite its higher reward. In terms of realized safety, our ablation points the opposite way to the concern that stochasticity increases out-of-distribution risk: in this evaluation, the deterministic variant incurs substantially higher realized costs.

By contrast, the entropy bonus acts only as a mild regularizer. Disabling it while keeping the stochastic policy leaves the policy within the safe region and changes the trade-off only modestly ($0.704/0.659 \rightarrow 0.669/0.694$). Safety-related behavior is instead controlled primarily by cost conditioning and the feasibility-reweighted prompt context.

Table 15: Computational overhead on the Car four-task setting (decision context length 20, prompt length 20, RTX 6000 Ada). CDT does not use a prompt. The per-action value divides the total time of a fixed batch by the batch size.

| Method | Train time/step (s) | Peak memory (GB) | Inference latency/action (ms) |
|---|---|---|---|
| Prompt-DT | 0.152 | 0.745 | 2.163 |
| CDT | 0.112 | 0.504 | 1.270 |
| GeoDT, uncached | 0.163 | 0.831 | 2.974 |
| GeoDT, cached | 0.163 | 0.831 | 2.129 |

### E.7 Computational Overhead and Caching

Constructing the structured context $c_{\text{geo}}$ introduces a one-time cost per prompt. When the prompt remains fixed during a rollout, $c_{\text{geo}}$ is computed once and cached, and recomputed only when the prompt changes; each decision step then reuses the cached context. Table 15 reports timing on the Car four-task setting with context length $K = 20$ and prompt length 20, measured on an RTX 6000 Ada GPU.

GeoDT requires 0.163 s per training step, about 7.2% more than Prompt-DT (0.152 s). Recomputing the context at every step yields 2.974 ms per action; caching reduces this by 28.4% to 2.129 ms, comparable to Prompt-DT (2.163 ms). CDT is faster (1.270 ms) because it performs no prompt processing. Overall, prompt-context construction adds a moderate one-time cost, while the practical per-step overhead is small once the context is cached.

### E.8 Prototype Retrieval Analysis

To examine how the shared prototype memory is used, we analyze prototype retrieval across tasks on the Car setting (Figure 7, retrieval-weight heatmap; Figure 8, retrieved trajectories). The retrieval pattern is inconsistent with a strict one-task/one-prototype lookup: it shows a mixture of cross-task reuse and task specialization. If each prototype simply encoded a task label, the usage matrix would be block-diagonal, with each task activating a single dedicated prototype. Instead, several prototypes are retrieved across tasks (e.g., $P_0$ and $P_6$ by Circle and Run, $P_4$ by Circle, Gather, and Run), while others are task-specialized (e.g., $P_1/P_7$ for Reach and $P_3/P_5$ for Gather).

The retrieved trajectories make the specialized prototypes interpretable: each consistently retrieves the characteristic motion of its task (e.g., $P_3$ retrieves Gather traversals; $P_1$ retrieves Reach goal-approaching paths). Among the shared prototypes, some groupings are visible in the $xy$ paths: $P_2$ retrieves near-linear, low-curvature motions common to Run and Reach, and $P_4$ retrieves curved, turning motions common to Circle and Gather. For $P_0/P_6$, the retrieved Circle and Run paths differ in $xy$ shape; since the transition-related branch is seven-dimensional (position, velocity, orientation, angular rate), such prototypes may group trajectories by velocity- or orientation-level structure not apparent in the $xy$ projection. We therefore ground the cross-task-sharing claim in the retrieval statistics, which directly show prototype reuse across tasks, rather than in visual path similarity alone. Together, these observations indicate that the prototype memory functions as a reusable, structure-organized memory containing both shared and task-specialized prototypes, rather than an implicit task-identifier mechanism.

## F Single-task performance

We report the single-task performance of all methods on four agents for reference in Table 16, 17, 18 and 19. Although GeoDT is not the best-performing model in every individual task—several baselines achieve top scores on specific tasks—it consistently delivers competitive results across all settings. As previously discussed, GeoDT exhibits strong robustness under increasing task diversity: as the number of tasks increases, it maintains performance close to the single-task upper bound, demonstrating its robustness and adaptability in multi-task environments.

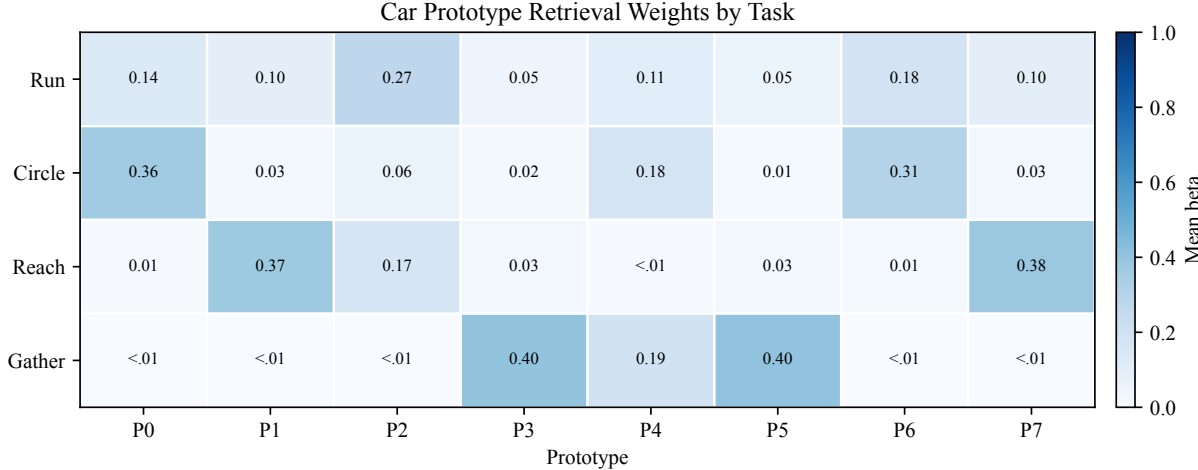

Figure 7: Prototype retrieval weights by task (Car). Each cell is the mean retrieval weight $\beta$ assigned to prototype P$_m$ by prompts from a given task. The pattern is not block-diagonal: several prototypes are retrieved by multiple tasks, while others are task-specialized, indicating a reusable structure-organized memory rather than a task-identity lookup table.

Table 16: Single-Task Performance on `Car` agent.

| Methods | Car-Run | | Car-Circle | | Car-Reach | | Car-Gather | | Average | |
|---|---|---|---|---|---|---|---|---|---|---|
| | Reward | Cost | Reward | Cost | Reward | Cost | Reward | Cost | Reward | Cost |
| GeoDT | 0.95 | 0.71 | 0.74 | 0.77 | 0.70 | 0.57 | 0.37 | 0.58 | 0.690 | 0.658 |
| CDT | 0.95 | 0.74 | 0.70 | 0.60 | 0.67 | 0.88 | 0.48 | 0.51 | 0.700 | 0.683 |
| PDT | 0.93 | 1.53 | 0.68 | 1.26 | 0.67 | 0.90 | 0.38 | 0.50 | 0.665 | 1.048 |
| CPQ | 0.75 | 0.52 | 0.65 | 0.53 | 0.55 | 1.33 | 0.15 | 0.55 | 0.525 | 0.733 |
| BCQL | 0.93 | 0.70 | 0.64 | 1.80 | 0.61 | 0.64 | 0.30 | 0.52 | 0.620 | 0.915 |
| LSPC | 0.71 | 0.22 | 0.78 | 0.23 | 0.71 | 0.73 | 0.48 | 0.55 | 0.671 | 0.430 |
| SoftMod | 0.93 | 0.64 | 0.75 | 2.56 | 0.70 | 0.70 | 0.38 | 0.70 | 0.690 | 1.150 |
| FISOR | 0.76 | 0.05 | 0.40 | 0.19 | 0.54 | 0.48 | 0.47 | 0.38 | 0.543 | 0.275 |
| O3SRL | 0.94 | 0.00 | 0.76 | 0.00 | 0.57 | 0.14 | 0.38 | 0.50 | 0.603 | 0.160 |

Table 17: Single-Task Performance on `Ball` agent.

| Methods | Ball-Run | | Ball-Circle | | Ball-Reach | | Ball-Gather | | Average | |
|---|---|---|---|---|---|---|---|---|---|---|
| | Reward | Cost | Reward | Cost | Reward | Cost | Reward | Cost | Reward | Cost |
| GeoDT | 0.79 | 0.84 | 0.77 | 0.63 | 0.89 | 0.84 | 0.48 | 0.55 | 0.733 | 0.715 |
| CDT | 0.88 | 0.86 | 0.79 | 0.82 | 0.88 | 0.64 | 0.45 | 0.55 | 0.750 | 0.718 |
| PDT | 0.79 | 0.81 | 0.73 | 0.90 | 0.72 | 0.75 | 0.49 | 0.43 | 0.683 | 0.723 |
| CPQ | 0.56 | 0.68 | 0.68 | 0.23 | 0.59 | 0.51 | 0.24 | 0.26 | 0.518 | 0.420 |
| BCQL | 0.35 | 0.20 | 0.79 | 1.20 | 0.88 | 0.75 | 0.46 | 0.52 | 0.620 | 0.668 |
| LSPC | 0.75 | 0.84 | 0.48 | 0.24 | 0.48 | 0.62 | 0.38 | 0.40 | 0.522 | 0.526 |
| SoftMod | 1.00 | 1.04 | 0.99 | 1.99 | 0.85 | 0.72 | 0.50 | 0.35 | 0.835 | 1.025 |
| FISOR | 0.31 | 0.00 | 0.44 | 0.01 | 0.65 | 0.78 | 0.44 | 0.73 | 0.460 | 0.380 |
| O3SRL | 0.66 | 0.02 | 0.63 | 0.02 | 0.88 | 0.94 | 0.37 | 0.50 | 0.633 | 0.369 |

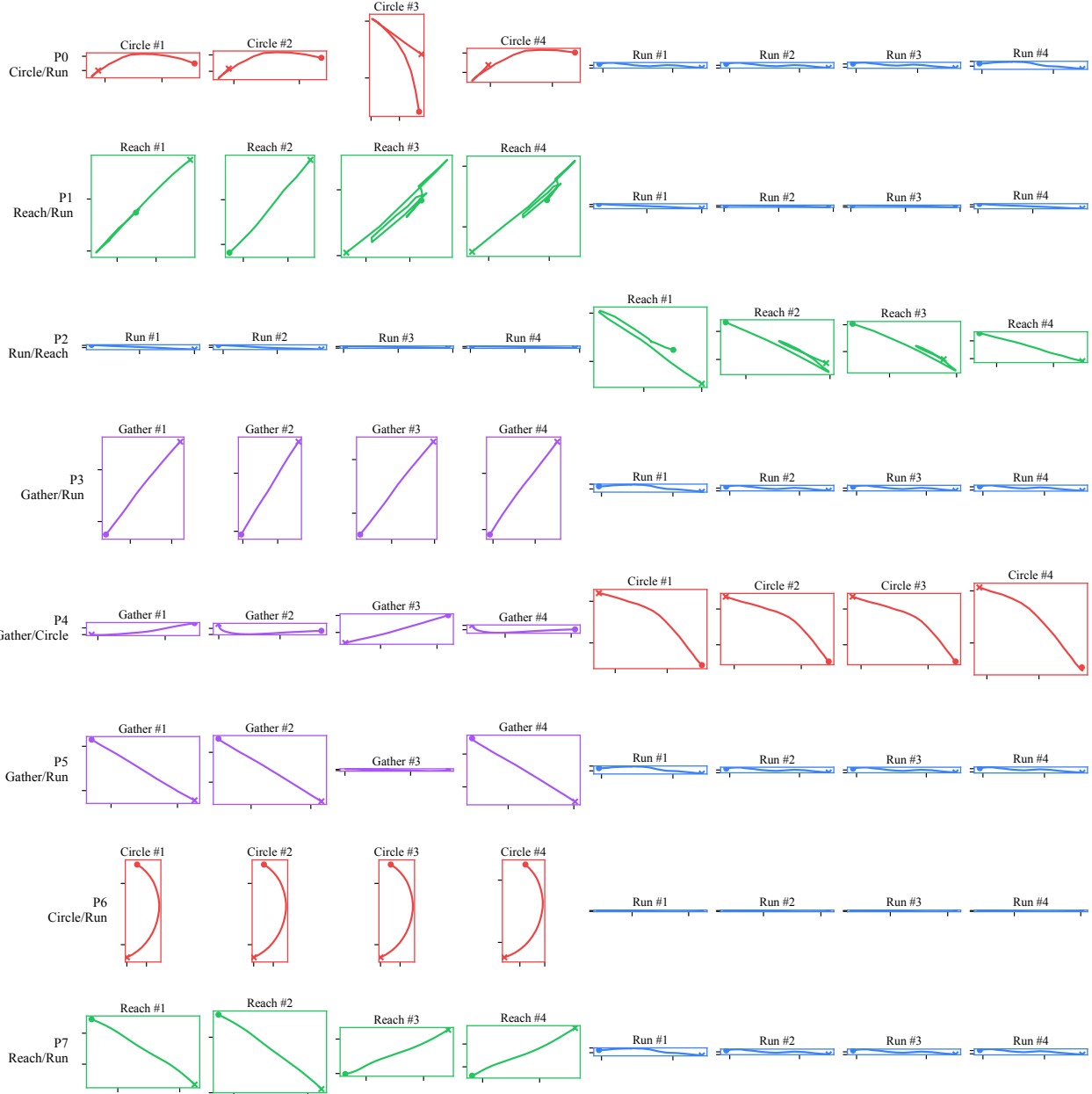

Figure 8: Trajectories retrieved by each prototype (Car), shown as $xy$ paths. Specialized prototypes (P1/P3/P5/P7) are dominated by one task's characteristic motion; shared prototypes (P0/P2/P4/P6) are retrieved across tasks.

Table 18: Single-Task Performance on `Drone` agent.

| Methods | Drone-Run | | Drone-Circle | | Drone-Reach | | Drone-Gather | | Average | |
|---|---|---|---|---|---|---|---|---|---|---|
| | Reward | Cost | Reward | Cost | Reward | Cost | Reward | Cost | Reward | Cost |
| GeoDT | 0.72 | 0.79 | 0.77 | 0.87 | 0.32 | 0.78 | 0.11 | 0.30 | 0.480 | 0.685 |
| CDT | 0.71 | 0.60 | 1.14 | 0.78 | 0.46 | 1.27 | 0.15 | 0.30 | 0.615 | 0.738 |
| PDT | 0.54 | 0.36 | 0.42 | 0.83 | 0.20 | 1.16 | 0.12 | 0.44 | 0.320 | 0.698 |
| CPQ | 0.26 | 0.44 | 0.56 | 0.56 | 0.15 | 0.53 | 0.06 | 0.08 | 0.258 | 0.403 |
| BCQL | 0.65 | 0.71 | 0.68 | 1.19 | 0.36 | 1.07 | 0.13 | 0.05 | 0.455 | 0.755 |
| LSPC | 0.72 | 0.69 | 0.66 | 0.67 | 0.45 | 0.90 | 0.25 | 0.25 | 0.518 | 0.627 |
| SoftMod | 0.56 | 0.23 | 0.96 | 1.95 | 0.30 | 0.77 | 0.13 | 0.25 | 0.488 | 0.800 |
| FISOR | 0.60 | 0.00 | 0.41 | 0.35 | 0.36 | 0.66 | 0.11 | 0.28 | 0.370 | 0.323 |
| O3SRL | 0.38 | 0.00 | 0.56 | 0.74 | 0.44 | 0.66 | 0.60 | 0.10 | 0.495 | 0.375 |

Table 19: Single-Task Performance on `Ant` agent.

| Methods | Ant-Run | | Ant-Circle | | Ant-Reach | | Ant-Gather | | Average | |
|---|---|---|---|---|---|---|---|---|---|---|
| | Reward | Cost | Reward | Cost | Reward | Cost | Reward | Cost | Reward | Cost |
| GeoDT | 0.90 | 0.43 | 0.44 | 0.74 | 0.24 | 0.68 | 0.20 | 0.05 | 0.445 | 0.475 |
| CDT | 0.92 | 0.62 | 0.85 | 1.34 | 0.14 | 1.03 | −0.02 | 0.06 | 0.472 | 0.763 |
| PDT | 0.38 | 0.21 | 0.10 | 0.18 | 0.07 | 0.85 | −0.08 | 0.02 | 0.118 | 0.315 |
| CPQ | 0.13 | 0.01 | 0.00 | 0.00 | 0.04 | 0.83 | −0.09 | 0.06 | 0.020 | 0.225 |
| BCQL | 1.02 | 4.63 | 0.61 | 1.42 | 0.17 | 0.72 | −0.13 | 0.02 | 0.418 | 1.698 |
| LSPC | 0.94 | 1.46 | 0.40 | 0.78 | 0.23 | 0.18 | 0.20 | 0.10 | 0.440 | 0.630 |
| SoftMod | 1.21 | 8.34 | 0.74 | 2.48 | 0.15 | 0.41 | 0.10 | 0.10 | 0.550 | 2.833 |
| FISOR | 0.60 | 0.38 | 0.41 | 0.00 | 0.18 | 0.29 | 0.15 | 0.05 | 0.335 | 0.180 |
| O3SRL | 0.37 | 0.54 | 0.50 | 0.00 | 0.20 | 0.81 | 0.30 | 0.00 | 0.343 | 0.338 |

