# OpenReview forum: "GeoDT: Geometry-Inspired Decision Transformer for Robust Safe Multi-Task Offline Reinforcement Learning"
_TMLR — Decision pending for TMLR_

### Review · Reviewer_aVWL · 2026-05-25

**Summary Of Contributions:**

This paper proposes GeoDT, a geometry-aware Decision Transformer for safe multi-task offline reinforcement learning. Its main contribution is to mitigate cross-task semantic inconsistency by separating geometry-related trajectory structure from task-specific semantics, while using cost signals to guide safe geometric reuse.

**Audience:**

Yes

**Audience Explanation:**

The ideas and experimental results presented in this paper are quite promising, and I believe that if the authors can provide more compelling evidence, it will attract some attention within the community.

**Claims And Evidence:**

No

**Claims Explanation:**

A major weakness is that the paper does not convincingly validate its central “geometry-aware” claim. The proposed geometry–semantics decomposition is learned through a highly flexible gating mechanism, yet the paper provides limited evidence that the resulting geometric branch actually captures reusable geometric or dynamical structure, rather than simply learning another generic latent representation.
The empirical support is also largely indirect.

**Requested Changes:**

The main need is to provide more direct evidence regarding the issues raised in the "Weakness" section. Additionally, some other articles using geometric features (though perhaps in different directions) should be cited below, but the motivation and evidence for using the term "geometry" must be explained.

[A] G-llava: Solving geometric problem with multi-modal large language model. ICLR'25.

[B] Large language-geometry model: When llm meets equivariance. ICML'25.

[C] Geogpt4v: Towards geometric multi-modal large language models with geometric image generation. EMNLP'24.

---

> ### Author Response · Authors · 2026-06-22
>
> We thank the reviewer for the encouraging assessment and the concrete request for more direct evidence. The reviewer's central concern is whether the learned decomposition produces a meaningful transition-related representation or merely another generic latent, and whether the term "geometry'' is adequately motivated. We first clarify the intended geometry-inspired interpretation and then present direct evidence supporting it.
>
> **What "geometry'' means in this work.**
> We agree that the original presentation did not clearly distinguish our use of "geometry'' from formal geometric reasoning or equivariant modeling. We have therefore clarified and more carefully positioned the claim throughout the revised manuscript. Here, "geometry'' refers operationally to the organization and evolution of agent-centric spatial and motion variables along trajectories, together with their action-conditioned transition regularities. The relevant representation includes variables such as position, velocity, orientation, and angular motion and focuses on how these variables form temporally related trajectory structure that can be reused across heterogeneous tasks.
>
> This interpretation differs from the suggested references, which we now discuss in Related Work (Section 2). G-LLaVA and GeoGPT4V study explicit geometric problem solving through geometric diagrams and multimodal reasoning, while EquiLLM incorporates $E(3)$-equivariant structure into the modeling of three-dimensional physical systems. GeoDT does not claim Euclidean constraints, equivariance, or symbolic geometric reasoning. Instead, it uses a geometry-inspired, agent-centric trajectory bias. We have therefore changed the title to *Geometry-Inspired Decision Transformer*, and the label "geo'' denotes this intended inductive bias rather than a formal geometric guarantee.
>
> **Why the transition-related branch is not an unconstrained generic latent.**
> The branch is constrained by both its construction and its functional role. First, a state-dependent soft gate explicitly routes input information into the transition-related and complementary branches, rather than mapping the entire state into an arbitrary undifferentiated embedding. Second, the one-step consistency objective provides an action-conditioned, transition-predictive criterion for shaping the former branch. Third, the branches are used asymmetrically: the transition-related branch is the sole input to within-prompt relational induction, feasibility-aware reweighting, and shared prototype retrieval, while the complementary branch directly preserves task-dependent information for policy conditioning. We do not claim that this produces an information-pure partition, but it gives the two branches distinct optimization biases and functional roles.

---

> > ### Author Response · Authors · 2026-06-22
> >
> > **Direct representation-level evidence.**
> > We added three diagnostics on the four-task Car and Ball settings in Section 5.5.
> >
> > *(i) Branch probing* (Table 3). We froze the trained representations and fit linear probes on held-out transitions. The probes evaluate next-step agent-centric dynamics predictability and the predictability of current task-dependent observation variables. The transition-related branch $z_t^{\mathrm{geo}}$ and the complementary branch $z_t^{\mathrm{sem}}$ exhibit a clear crossover rather than one branch being uniformly stronger. On Car, $z_t^{\mathrm{geo}}$ retains nearly all of the dynamics predictability of the no-decomposition representation ($R^2=0.969$ versus $0.981$), while its task-dependent predictability is substantially lower ($0.781$ versus $0.953$). In contrast, $z_t^{\mathrm{sem}}$ preserves task-dependent predictability ($0.941$) but is markedly weaker for dynamics ($0.836$). Ball exhibits the same pattern. This crossover supports preferential branch specialization rather than two undifferentiated generic representations.
> >
> > *(ii) Gate routing* (Figure 4). We directly inspected the learned input-level gate by averaging its values over agent-centric and task-dependent observation dimensions. Agent-centric dimensions consistently receive larger gate weights toward the transition-related branch across all tasks with an explicit task-dependent block (mean difference ranging from $0.13$ to $0.33$ on Car and $0.20$ to $0.34$ on Ball). Run is omitted because it lacks an explicit task-dependent observation block. These results show that the learned routing preference is consistent with the intended agent-centric inductive bias.
> >
> > *(iii) Learned versus manual decomposition* (Table 4). We compared the learned gate with a fixed, task-agnostic split that assigns predefined agent-centric dimensions to the transition-related branch and the remaining dimensions to the complementary branch. The result shows that the manual split is not a reliable substitute: it raises overall normalized cost from $0.659$ to $1.440$ while lowering reward from $0.704$ to $0.646$. This result suggests that the fixed binary grouping is insufficient and that the learned soft, state-dependent allocation better supports the reward-cost trade-off.
> >
> > Together, the mechanism design, probing crossover, gate statistics, and manual-split comparison support the interpretation that the learned branch preferentially retains agent-centric, transition-related trajectory information. They do not establish an exact geometric factorization, and we have revised the manuscript to make this scope explicit.

---

### Review · Reviewer_YJob · 2026-06-04

**Summary Of Contributions:**

This paper proposes GeoDT, a geometry-aware Decision Transformer for safe multi-task offline RL. The method builds upon both Prompt-DT-style prompting and cost-conditioned Decision Transformer frameworks, while introducing a learned decomposition of states into "geometry" and "semantic" components and a geometry-aware prompt context that is integrated into the policy.

The empirical evaluation is fairly extensive and shows improved reward-cost trade-offs over several baselines. However, the method is quite complex, and I am not fully convinced by the geometry-semantics interpretation. The "geometry" component is mainly encouraged through a self-predictive dynamics loss, while the "semantic" component is defined largely as its complement. As a result, the method appears stronger as an extension of prompt- and cost-conditioned sequence modeling than as a demonstrated geometry-aware representation learning approach.

**Additional Comments:**

I have couple of following questions to the authors:

1. The prototype memory bank appears to be queried during inference to construct the geometry-aware context. Could the authors clarify how $c^{\mathrm{geo}}$ is obtained at inference time? In particular, is it recomputed at every decision step or cached when the prompt remains fixed? A discussion of the computational overhead relative to Prompt-DT or cost-aware Decision Transformer baselines would also be helpful.

2. Since the benchmark observations appear to contain identifiable agent-centric and task-dependent components, have the authors considered comparing against a manually specified geometry/semantic decomposition? Such a comparison could help assess whether the learned mask discovers a meaningful structure.

3. The prototype memory is described as capturing geometric structure, but the interpretation remains somewhat implicit. Is there any qualitative evidence supporting this claim, such as visualizations of prototypes or examples of associated trajectories?

**Audience:**

Yes

**Audience Explanation:**

Yes.

The paper addresses safe BC/offline RL, multi-task RL, and Decision Transformer-based policy learning, all of which are active research areas. Researchers working on prompt-conditioned RL and safe policy learning would likely find the results relevant.

That said, the contribution is primarily a combination of several existing ideas, and some design choices are not fully justified. Improving the clarity of the method and its positioning would strengthen the paper.

**Broader Impact Concerns:**

No major concerns. The paper studies safe BC/offline RL and may help reduce unsafe online deployment.

**Claims And Evidence:**

Yes

**Claims Explanation:**

Mostly yes.

The experimental results support the claim that GeoDT improves performance on the evaluated safe multi-task offline RL benchmarks. The ablations also suggest that the proposed components contribute to the final performance.

My main concern is with the interpretation of the learned decomposition. The self-predictive objective encourages $s^{\mathrm{geo}}$ to capture transition-relevant information, but it is not clear that this corresponds to geometric information. Likewise, $s^{\mathrm{sem}}$ is not explicitly encouraged to encode task semantics. I therefore think the empirical performance claims are supported, but the representation-learning claims should be stated more carefully.

**Requested Changes:**

### Improve the presentation and overall coherence of the method section

The method section is difficult to follow because the framework contains many interacting components (state decomposition, geometry consistency learning, prompt attention, safety-aware reweighting, prototype retrieval, and cost-conditioned action prediction). I would strongly encourage the authors to first present the complete pipeline in a compact end-to-end form using general mathematical notation, ideally when introducing Figure 1, before discussing each module separately.

I found the training algorithm in the appendix helpful. Moving it to the main paper would improve readability and clarify how the components interact. The appendix also lacks an inference algorithm. Including one would help readers understand how the geometry-aware context is constructed and used at deployment.

---

### Clarify the relation to Prompt-DT and prior DT variants

The method appears heavily inspired by Prompt-DT and cost-conditioned Decision Transformers, but the distinction from these approaches is not always clear. The preliminaries mainly discuss the original Decision Transformer, while Prompt-DT receives relatively little discussion despite being closely related to the proposed setting.

I would recommend adding a brief discussion of Prompt-DT (or prompt-conditioned multi-task DTs more generally) in the preliminaries and explicitly highlighting which components are inherited from prior work and which components are newly introduced in GeoDT.

---

### Clarify what is meant by "geometry"

This is my main concern. The method does not appear to use explicit geometric structure; instead, the geometry branch is encouraged through a one-step predictive loss. As a result, the learned representation seems more naturally interpreted as a transition-predictive or dynamics-related component rather than necessarily a geometric one.

It would be helpful to either better justify the terminology or soften the interpretation throughout the paper. A comparison against a manually specified geometry/semantic split (if feasible in these environments) would also strengthen the claim and help validate whether the learned decomposition captures the intended structure.

---

### Clarify the relational attention module

In Eqs. (8)-(9), please explicitly define what the indices $i$ and $j$ refer to. It is currently unclear whether the attention is computed over states within a prompt trajectory, across multiple prompts, or across tasks. A brief clarification would be sufficient and would help readers better understand how the attention mechanism relates to the claimed cross-task context construction.

---

### Justify the role of the entropy term

The motivation for the entropy term is unclear. In an offline BC/RL setting, why do we need stochastic policies and increased entropy? Higher entropy could increase the likelihood of out-of-distribution actions, so the authors should explain why it is needed and ideally provide an ablation study or sensitivity analysis.

---

> ### Author Response · Authors · 2026-06-22
>
> We thank the reviewer for the constructive and insightful feedback. Specifically, in the revision, we have improved the presentation of the proposed GeoDT pipeline, added explicit training and inference procedures, clarified GeoDT's relationship to Prompt-DT and prior multi-task Decision Transformer variants, clarified and more carefully positioned the geometry-inspired interpretation, specified the scope of the within-prompt relational attention, and added targeted analyses of the stochastic policy, entropy regularization, inference-time caching, learned versus manual decomposition, and prototype retrieval. We respond to each point below and indicate where the corresponding changes and evidence appear in the revised manuscript.
>
> **Method presentation (compact pipeline and algorithms).**
> We agree that the original presentation made the interaction among the components unnecessarily difficult to follow. In the revision, we have added an end-to-end description of the full pipeline in Section 4. We also moved the descriptions for the training procedure into the main text and added an inference algorithm in Appendix C.3. The inference procedure explicitly shows that the structured prompt context $c_{\mathrm{geo}}$ is constructed once and cached when the prompt remains fixed, while subsequent action predictions reuse the cached context.
>
> **Related Work Discussion.**
> We thank the reviewer for suggesting a broader related-work discussion. In the revised version, we have expanded both the Preliminaries (Section 3.2) and Related Work (Section 2) to position GeoDT more clearly among prior Decision Transformer (DT) variants. DT formulates offline RL as a conditional sequence modeling problem, while constrained or safe DT variants such as CDT and SaFormer extend this formulation with return-cost conditioning. These methods provide the sequence-modeling backbone and safety-conditioning mechanisms inherited by GeoDT, but primarily focus on single-task or task-aligned settings and do not directly address semantic inconsistency under shared heterogeneous multi-task representations.
>
> We also discussed multi-task DT methods that use explicit task information or task-aware parameterization. For example, some multi-task DT baselines include task-ID embeddings in the Transformer input, HarmoDT learns task-specific parameter masks and subspaces to mitigate multi-task conflicts, and M3DT uses task grouping and expert specialization for large-scale multi-task learning. CoPDT considers constraint-prioritized prompt encoding under varying safety budgets; however, in its cross-environment setting, it uses environment-specific state and action encoders selected according to an environment identity. These methods demonstrate that task-aware routing or specialization can improve multi-task sequence modeling, although dependence on explicit task or environment information may restrict their use when such identity is unavailable at deployment.
>
> Prompt-DT is the closest predecessor to GeoDT because it uses trajectory prompts as implicit task or context descriptors rather than explicit task IDs. GeoDT inherits this prompt-conditioned view, but differs in both setting and mechanism. Prompt-DT directly conditions on raw trajectory prompts to support few-shot adaptation to held-out tasks within related task families, whereas GeoDT targets identifier-free joint training over semantically heterogeneous safe tasks. Instead of using the raw prompt directly as a generic context, GeoDT constructs a structured context $c_{\mathrm{geo}}$ through geometry-inspired decomposition, relational structure induction, feasibility-aware reweighting, and prototype retrieval, and then fuses it with complementary task-dependent state features in a cost-aware DT. Empirically, Prompt-DT under our protocol achieves a competitive reward but a substantially higher normalized cost of 1.414 (Table 1). This indicates that raw-prompt conditioning does not match GeoDT's safety performance in our joint multi-task setting.

---

> > ### Author Response · Authors · 2026-06-22
> >
> > **Clarifying "geometry''.**
> >
> > We agree that the original presentation did not clearly distinguish our operational use of "geometry'' from formal geometric or equivariant modeling. We have therefore clarified and more carefully positioned the claim throughout the revised manuscript. In this work, "geometry'' refers to the organization and evolution of agent-centric spatial and motion variables along trajectories, together with their action-conditioned transition regularities. This includes information such as position, velocity, orientation, and angular motion, but the intended notion is not limited to individual coordinates; it concerns how these variables form temporally related trajectory structure that can be reused across tasks.
> >
> > The resulting transition-related branch is also more constrained than a generic latent representation. First, it is produced through a state-dependent soft routing mechanism that explicitly determines which input information enters the branch. Second, the one-step consistency objective biases it toward action-conditioned, transition-predictive information. Third, the two branches have asymmetric functional roles: the transition-related branch is used to induce within-prompt relations, apply feasibility-aware reweighting, and retrieve shared prototypes, whereas the complementary branch preserves task-dependent information for policy conditioning.
> >
> > We add direct evidence for this interpretation in Section 5.5. Linear probing shows a clear crossover: the transition-related branch preserves more next-step agent-centric dynamics information, whereas the complementary branch preserves more task-dependent information (Table 3). The learned gate also assigns consistently larger average weights to agent-centric observation dimensions (Figure 4). Finally, replacing the learned gate with a fixed manual split lowers overall reward from $0.704$ to $0.646$ and raises normalized cost from $0.659$ to $1.440$ (Table 4). Together, these results support preferential branch selectivity rather than an unconstrained generic latent or an information-pure disentanglement.
> >
> > Accordingly, we have changed the title to *Geometry-Inspired Decision Transformer*. GeoDT does not impose Euclidean constraints, equivariance, or symbolic geometric reasoning; the name and the label "geo'' denote the intended agent-centric, transition-related inductive bias rather than a formal geometric guarantee.
> >
> > **Clarification of the relational attention module.**
> > In the revised version, we also explicitly defined the attention indices in Section 4.2. The indices $i,j$ in the original version are written as $u,v$ in the revision and refer to the positions of the timesteps within the same prompt trajectory: $u,v\in$ {1, ..., K }. Relational attention is therefore computed within a single prompt rather than across prompts or tasks. Cross-task sharing arises from both the model parameters jointly learned across all task datasets and the shared prototype memory, which is queried by each prompt summary.

---

> > > ### Author Response · Authors · 2026-06-22
> > >
> > > **Role of the stochastic policy and entropy term.**
> > > We clarified that the stochastic Gaussian parameterization and the entropy bonus are two separate design choices. The entropy bonus is not intended to encourage arbitrary exploration or out-of-distribution actions. We followed the stochastic policy design of CDT/ODT-style Decision Transformers, using a conditional Gaussian policy rather than deterministic action prediction. CDT motivates this choice in offline safe RL by noting that deterministic policies can be brittle under extrapolation error and may commit to unsafe or unsupported actions, and reports ablations supporting the stochastic formulation. This consideration is particularly relevant in our multi-task prompt-conditioned setting: similar states or prompt contexts may correspond to multiple valid actions because of differences in task objectives, target cost budgets, or behaviour modes in the offline data. A stochastic policy can represent this conditional action uncertainty, while the likelihood objective anchors the learned distribution to actions observed in the offline dataset.
> > >
> > > Table 14 separately ablates the stochastic parameterization and the entropy bonus under the same evaluation protocol. Our results suggest that the stochastic parameterization contributes more strongly to the observed reward-cost trade-off than the entropy bonus. Replacing the stochastic policy with a deterministic action predictor and retraining the model accordingly increases overall normalized cost from $0.659$ to $1.620$, placing the policy outside the safe region despite a higher reward. Thus, in our evaluation, the deterministic variant incurs substantially larger realized safety violations. By contrast, removing only the entropy bonus while retaining the stochastic Gaussian policy produces a much smaller change: $0.704/0.659 \rightarrow 0.669/0.694$ in overall reward/cost, and the policy remains within the safe region. We therefore interpret the entropy term as a mild regularizer rather than the primary safety mechanism. In our design, safety is addressed primarily through cost conditioning and the feasibility-aware structured context; the entropy term serves as a secondary regularizer. The complete comparison is reported in Appendix E.6.

---

> ### Author Response · Authors · 2026-06-22
>
> **Question 1 (inference-time cost of $c_{\mathrm{geo}}$).**
> We thank the reviewer for raising this question. At inference time, $c_{\mathrm{geo}}$ is constructed from the demonstration prompt through learned decomposition, relational aggregation, feasibility weighting, and prototype retrieval. When the prompt remains fixed during a rollout, this context is computed once and cached; it is recomputed only when the prompt changes. Each subsequent decision step reuses the cached $c_{\mathrm{geo}}$ together with the current policy context.
>
> We also measured the computational overhead in the same Car four-task setting, with decision context length $20$ and prompt length $20$, on an RTX 6000 Ada GPU (Appendix E.7, Table 15). GeoDT requires $0.163$ s per training step, approximately $7.2\%$ more than Prompt-DT at $0.152$ s. Recomputing the structured prompt context at every step results in $2.974$ ms per action, whereas caching it reduces latency by $28.4\%$ to $2.129$ ms, which is comparable to Prompt-DT at $2.163$ ms. CDT is faster at $1.270$ ms because it does not perform prompt processing. These results show that prompt-context construction introduces a moderate one-time cost, while the practical per-step overhead is small when the prompt is cached.
>
> **Question 2 (manual versus learned decomposition).**
> Thank you for the question. We added the requested comparison in Section 5.5 (Table 4). The manual variant uses a fixed, task-agnostic binary mask that assigns predefined agent-centric dimensions to the transition-related branch and all remaining task-dependent dimensions to the complementary branch. This fixed split is not a reliable substitute for learned decomposition: overall reward decreases from $0.704$ to $0.646$, while normalized cost increases from $0.659$ to $1.440$. For the Ball task, reward/cost changes from $0.710/0.698$ to $0.579/1.593$. For the Car task, the manual split obtains slightly higher reward ($0.712$ versus $0.697$), but only by substantially increasing cost ($0.620$ to $1.286$). These results suggest that the fixed binary grouping is insufficient and that the learned soft, state-dependent allocation better supports the reward-cost trade-off.
>
> **Question 3 (prototype interpretation).**
> We added a prototype-retrieval analysis on the Car setting, including a retrieval-weight heatmap and representative retrieved trajectories (Appendix E.8, Figures 7 and 8). The retrieval matrix does not exhibit a strict one-task/one-prototype pattern. Several prototypes are reused across tasks (e.g., $\mathrm{P}0$ and $\mathrm{P}6$ by Circle and Run, and $\mathrm{P}4$ by Circle, Gather, and Run), while others are more task-specialized (e.g., $\mathrm{P}1/\mathrm{P}7$ for Reach and $\mathrm{P}3/\mathrm{P}5$ for Gather). The retrieved trajectories make several specialized prototypes directly interpretable: Reach prototypes retrieve goal-approaching paths, while Gather prototypes retrieve characteristic traversal patterns. Among the shared prototypes, some groupings are also visible in the $xy$ paths: $\mathrm{P}2$ retrieves near-linear, low-curvature motions from Run and Reach, while $\mathrm{P}4$ retrieves curved, turning motions from Circle and Gather. For some shared prototypes, however, the similarity is less apparent from the plotted $xy$ trajectories alone. This limitation is unsurprising because the visualization shows only a two-dimensional projection of the seven-dimensional agent-centric representation, which additionally includes velocity, orientation, and angular-rate variables. We therefore use the projected trajectories as complementary qualitative evidence and ground the cross-task-sharing claim primarily in the retrieval statistics. Overall, these results support interpreting the prototype memory as containing a mixture of cross-task-shared and task-specific prototypes, rather than functioning as a simple task-identifier lookup.

---

### Review · Reviewer_zcNw · 2026-06-11

**Summary Of Contributions:**

The paper studies the problem of offline safe Reinforcement Learning in the multiple tasks setting. The proposed method is a workflow with multiple moving parts that uses a learned gating mechanism to split states into a dynamics-aware component (regularized by forward state prediction) and a semantic component, which are then used to build a cost-reweighted trajectory context to condition a Decision Transformer. Experiments are performed on few standard benchmarks.

**Audience:**

No

**Audience Explanation:**

Neither the problem statement nor the ideas are convincing to be useful to the community.

**Claims And Evidence:**

No

**Claims Explanation:**

- There are many confusing and incorrect usage of terminology in the paper. The paper regularly uses terms like "geometry," "geometric regularities," and "geometry-semantics decomposition." However, there is nothing inherently geometric about the proposed method. The main term is a standard auxiliary dynamics prediction.

- The paper doesn't make a convincing case for studying the combined setting of offline safe RL with multiple tasks. The reasons describing relevance of prior work are a bit generic with vague terminology and without any technical depth  ("such approaches leave under-specified what structure should be shared across heterogeneous tasks", "primarily focus on performance generalization rather than semantic misalignment under joint training.").

- The proposed approach has multiple parts of the workflow GeoDT contains many moving parts including state masking, forward dynamics loss, self-attention relational matrices, cost-based reweighting, and a global prototype memory. Such workflows are typically brittle and have limited practical usage. The paper doesn't make provide a strong justification for all of these components. The ablation table 2 shows that the removal of the  L-geo consistency loss causes a larger drop in performance than removing the decomposition itself which may point to only the auxiliary dynamics loss being more important.

- Unfortunately the theory is a bit vaccous with no insights. This is not a major issues but makes the current form of the paper weak coupled with other big challenges of the paper.

**Requested Changes:**

Please see my concerns above.

---

> ### Author Response · Authors · 2026-06-22
>
> We thank the reviewer for the assessment. In response to the review, we have made five substantive revisions. We clarified and more carefully positioned the geometry-inspired claim throughout the manuscript, including its intended meaning, functional role, and empirical support (Concern 1); made the motivation for the combined safe, offline, and heterogeneous multi-task setting more concrete (Concern 2); added a controlled factorial ablation that directly tests whether a generic auxiliary prediction loss alone explains the gain (Concern 3); provided sensitivity and cross-embodiment evidence addressing the concern about brittleness (Concern 4); and repositioned the appendix analysis explicitly as illustrative rather than as a complete characterization of GeoDT's optimization dynamics (Concern 5).
>
> **Concern 1 (meaning of "geometry'' and the role of the representation).**
> We agree that the original presentation did not sufficiently distinguish our operational use of "geometry'' from formal geometric modeling. This lack of clarity could make the method appear either to claim Euclidean or equivariant structure that it does not impose, or to use "geometry'' merely as a label for a generic auxiliary latent. We have revised the manuscript to state the claim more precisely rather than simply retaining the original terminology.
>
> In GeoDT, "geometry'' refers to the organization and evolution of agent-centric spatial and motion variables along trajectories, together with their action-conditioned transition regularities. The relevant variables include position, velocity, orientation, and angular motion, but the intended structure is not any individual coordinate or a hand-crafted partition. Rather, it is the temporally related, action-conditioned trajectory structure formed by these variables that can provide a more stable basis for cross-task sharing than task-dependent goals, boundaries, obstacles, or sensor signals. This is why we use the more accurate term *geometry-inspired* and changed our title to *Geometry-Inspired Decision Transformer*. GeoDT does not impose Euclidean constraints, equivariance, symbolic geometric reasoning, or an information-pure geometric factorization.
>
> The proposed branch also differs from a generic latent representation in three concrete ways. First, it is created by a state-dependent soft routing mechanism that explicitly allocates input information between a transition-related branch and a complementary task-dependent branch. Second, the one-step consistency objective supplies an action-conditioned, transition-predictive criterion for shaping the former branch. Third, the two branches have distinct downstream roles: only the transition-related branch is used to induce within-prompt relations, apply feasibility-aware reweighting, and retrieve shared prototypes, whereas the complementary branch preserves task-dependent information for direct policy conditioning. Thus, $L_{geo}$ is a learning criterion within an asymmetric representation-and-context construction mechanism, rather than the complete contribution or a standalone dynamics regularizer.
>
> We add several direct tests of this interpretation. Branch probing shows that the transition-related branch preserves substantially more next-step agent-centric dynamics information, whereas the complementary branch preserves more task-dependent information (Section 5.5, Table 3). Gate statistics show that agent-centric dimensions receive consistently larger weights toward the transition-related branch (Figure 4). A fixed manual split does not reproduce the benefit of the learned soft routing and substantially worsens the reward-cost trade-off (Table 4). In addition, the $2{\times}2$ factorial ablation in Concern3 shows that applying $L_{geo}$ directly to an undifferentiated shared representation does not reproduce the benefit of combining the loss with the soft decomposition. The prototype-retrieval analysis further shows a mixture of cross-task reuse and task-specialized retrieval rather than a simple task-identity lookup (Appendix E. 8, Figures 7 and 8).
>
> Taken together, these results support a specific and limited claim: GeoDT learns a preferentially transition-related, agent-centric pathway that is functionally distinct from an unconstrained generic latent and serves as the basis for structured cross-task context construction. Accordingly, we have changed the title to *Geometry-Inspired Decision Transformer*, retained the name GeoDT and the label "geo'' to denote this inductive bias, and removed wording that could be interpreted as claiming formal geometric guarantees or exact disentanglement.

---

> ### Author Response · Authors · 2026-06-22
>
> **Concern 2 (motivation for the combined setting).**
> Thank you for your comment. In the revised manuscript, we have clarified the motivation in the Introduction and Related Work, particularly the interaction among task heterogeneity, fixed offline data, and safety constraints. Safe multi-task offline RL is not merely a direct juxtaposition of safe offline RL and multi-task RL: task heterogeneity, fixed offline data, and safety constraints interact in ways that produce a distinct failure mode.
>
> First, strong single-task safe offline RL performance does not transfer trivially to joint heterogeneous training. Under our protocol, FISOR obtains an average reward/cost of $0.543/0.275$ over the four single-task Car settings, but changes to $0.353/0.358$ under joint multi-task training. LSPC exhibits degradation in both metrics ($0.671/0.430 \rightarrow 0.430/0.838$), while O3SRL maintains a comparable reward, but its average cost increases from $0.160$ to $1.393$. Thus, the degradation caused by joint training can appear both as reduced task performance and as increased safety violations. This observation also motivates our TSRS and ITBS metrics, which quantify performance retention relative to single-task references and balance across jointly trained tasks.
>
> Second, the semantic inconsistency in our benchmark is concrete rather than purely abstract. As detailed in Appendix A, the observations contain a relatively stable agent-centric block, while the task-dependent portion changes meaning across environments. The Run task has no explicit task-dependent observation block; the Circle task introduces a boundary-distance signal; the Reach task includes goal-distance and LIDAR signals; and the Gather task includes object- and obstacle-related LIDAR signals. Consequently, aligned input dimensionality does not imply aligned semantics, and a jointly trained encoder can receive conflicting optimization signals from features occupying corresponding input positions but carrying different meanings.
>
> Third, reducing the dependence on explicit task identity is useful when task labels are unavailable, ambiguous, or require a separate identification mechanism. Existing multi-task sequence models commonly use task-aware mechanisms such as task-ID embeddings, task-specific parameter masks, expert grouping, or environment-specific encoders. GeoDT instead uses the trajectory prompt as its context signal and softly routes state information into a transition-related branch and a complementary task-dependent branch. We therefore replaced vague phrases such as "under-specified shared structure'' with the explicit failure mode studied here: directly sharing semantically mixed observation features can entangle reusable agent-centric transition patterns with task-dependent environmental cues, and under safety constraints, this negative transfer can manifest as both lower reward and higher cost.

---

> ### Author Response · Authors · 2026-06-22
>
> **Concern 3 (is the gain explained by an auxiliary prediction loss?).**
> In the revised version, we have addressed this question through a controlled $2{\times}2$ factorial experiment using the same joint four-task protocol (Section 5.4, Table 2). The four matched configurations use the same datasets, prompt construction, training budget, evaluation protocol, and common model settings, while varying whether the learned decomposition and $L_{geo}$ are present. A shared encoding without $L_{geo}$ obtains an overall reward/cost of $0.676/0.711$. Applying $L_{geo}$ directly to the same undifferentiated representation---the direct control for a generic auxiliary prediction loss---slightly reduces cost but also lowers reward, yielding $0.657/0.700$. It therefore does not produce a consistent reward-cost improvement. Conversely, using the decomposition without$L_{geo}$ yields $0.650/0.768$. Only their combination achieves both the highest reward and the lowest cost ($0.704/0.659$).
>
> These results indicate that the observed gain cannot be attributed to a generic auxiliary prediction objective alone. The policy objective by itself does not provide a transition-based criterion for learning the routing mask. $L_{geo}$ supplies such a criterion by biasing one branch toward action-conditioned, transition-predictive information. When applied directly to the undifferentiated representation in the shared-encoding control, however, it provides only a generic predictive regularization signal and does not reproduce the benefit of the complete method.
>
> The remaining components also contribute beyond $L_{geo}$. Removing feasibility-aware reweighting changes the overall reward/cost from $0.704/0.659$ to $0.639/0.868$, while removing prototype retrieval changes it to $0.635/0.737$ (Section 5.4, Table 2). The decomposition and the consistency objective shape the transition-related branch, whereas relational modeling, feasibility-aware reweighting, prototype retrieval, and complementary task-dependent conditioning determine how that branch is converted into a useful safe multi-task context. The full performance, therefore, cannot be attributed to $L_{geo}$ alone.
>
> **Concern 4 (many components and potential brittleness).**
> We do agree that there are several components in our proposed solution. However, we would like to respectfully point out that in the revised version, we have shown that our proposed solution is generally robust, i.e., it shows strong robustness across different settings with a single set of parameters. Specifically, we address this concern through sensitivity analysis, deployment cost, and cross-embodiment evaluations. Experiment results show that GeoDT remains relatively stable when the number of prototypes is varied over $M\in\{4,8,16\}$, and varying the prompt length over $\{10,20,40\}$ changes the reward-cost balance without causing performance collapse. These sensitivity results are reported in Appendix E. The same model design and common hyperparameter configuration are used across four substantially different embodiments---Car, Ball, Drone, and Ant---apart from environment-determined input and output dimensionalities. GeoDT maintains favorable reward-cost trade-offs across these settings without embodiment-specific redesign. In addition, the runtime analysis shows that the structured prompt context can be computed once and cached at deployment, making cached inference latency comparable to Prompt-DT. Together with the component ablations, these results suggest that the method is not narrowly tuned to a single prototype count, prompt length, embodiment, or deployment configuration.
>
> **Concern 5 (theoretical analysis).**
> We agree that the analysis in Appendix B is deliberately simplified and should not be interpreted as a complete theory of GeoDT. In the revision, we have retitled it as an illustrative analysis, foregrounded its assumptions, and stated explicitly that its purpose is only to formalize the intuition that task-dependent perturbations can produce conflicting gradients under shared heterogeneous training, while a transition-biased decomposition can reduce such sensitivity. This analysis is intended to provide supporting intuition rather than serve as the primary evidence for the method. The main evidential weight is instead carried by controlled factorial ablation, empirical gradient measurements, representation probing, gate-routing statistics, and manual split comparisons.